# Non-Monotonic Autoregressive Sequence Model

Tianyi Ma [1]  Yiyue Qian [2]  Yiyang Li [1]  Zehong Wang [1]  Yifan Ding [2]  Zheyuan Zhang [1]
Yan Liang [2]  Chuxu Zhang [3]  Yanfang Ye [1]

## Abstract

Autoregressive models generate sequences monotonically, where any sampled token, even if erroneous or sub-optimal, becomes a permanent condition for all subsequent steps. This structural limitation means that autoregressive models cannot revisit or revise earlier decisions, *i.e.*, a capability essential for complex generation tasks where exploration and correction are necessary. To this end, we propose **N-MARS**, a **N**on-**M**onotonic **A**uto**R**egressive **S**equence modeling framework that enables models to generate, evaluate, and revise tokens within a single forward pass, effectively allowing exploration before commitment. We operationalize this framework through a learned erase token $\langle \text{UNDO} \rangle$ that retracts the previous token, enabling on-the-fly revision within standard autoregressive decoding. To train the model, we introduce a sequence augmentation method that constructs error-correction trajectories from model-generated deviations paired with ground-truth references. We then propose masked supervised fine-tuning (mSFT) that exposes the model to errors as context for learning when to revise, without optimizing their likelihood. Afterward, we refine the model with group relative policy optimization (GRPO) that incentivizes judicious $\langle \text{UNDO} \rangle$ usage, *i.e.*, rewarding effective corrections while penalizing unsuccessful ones. We conduct comprehensive theoretical and empirical analysis to validate the effectiveness of N-MARS, demonstrating a robust foundation for non-monotonic sequence modeling. Our source code is available at https://github.com/Tianyi-Billy-Ma/N-MARS.

[1]University of Notre Dame [2]Amazon AI, The work is not related to the position(s) at the corresponding institution. [3]University of Connecticut. Correspondence to: Tianyi Ma <tma2@nd.edu>, Yanfang Ye <yye7@nd.edu>.

*Proceedings of the $43^{rd}$ International Conference on Machine Learning*, Seoul, South Korea. PMLR 306, 2026. Copyright 2026 by the author(s).

## 1. Introduction

Autoregressive models, such as large language models (LLMs) (Ye et al., 2025), have demonstrated excellent capabilities to perform various downstream tasks, from question-answering (Kamalloo et al., 2023; Song et al., 2025; Qian et al., 2026; 2025; Wang et al., 2026; Ye et al., 2026; Zhang et al., 2025b; Shi et al., 2026), code generation (Chen et al., 2021; Ma et al., 2025b), to complex summarizations (Zhang et al., 2024b; Tang et al., 2023; Li et al., 2025; Qian et al., 2026; Ma et al., 2025a). Fundamentally trained on the next-token prediction objective, these models may fail in complex generation tasks, due to their inherent monotonic generation process, where a new token is always appended to the immutable sequence (Bachmann & Nagarajan, 2024; Thoppilan et al., 2022). While this monotonic generation has become a fundamental paradigm, it is susceptible to error propagation, where a single incorrect token corrupts subsequent generations, leading to cascading failures that the model cannot recover from (Zhang et al., 2024a; Gan et al., 2025; Berglund et al., 2024; Zhu et al., 2024).

To address these limitations, recent research has explored methods to induce refinement capabilities (Lin et al., 2023) in LLMs, such as prompting strategies (Wei et al., 2022; Zhou et al., 2023; Zhang et al., 2026; Huang et al., 2026), search-based methods (Yao et al., 2023; Besta et al., 2024), and post-hoc verifications (Madaan et al., 2023; Lightman et al., 2024; Wang et al., 2023). While effective, these approaches operate outside the generation process, relying on expensive external verifiers or multiple inference passes, making them computationally inefficient. For example, several studies (Luo et al., 2024; Lightman et al., 2024) leverage a large-scale judger to validate output and trigger regeneration. Moreover, even with reinforcement learning (Huang et al., 2024; Kamalloo et al., 2023; Kamoi et al., 2024) for extended self-correction capabilities, these approaches offer no intrinsic signal of model confidence during generation, making it difficult to assess whether the model is on track or prone to deviating toward incorrect conclusions.

This naturally raises a research question: *Can we design an autoregressive sequence model that internally detects and corrects its errors during generation?*

To this end, we propose N-MARS, a novel **N**on-**M**onotonic

**Auto**Regressive **S**equence model, that internalizes the iterative refinement process directly into the autoregressive generation loop. Instead of relying on external mechanisms or post-hoc critique, we train the model with an erase action, which serves as a token $\langle$UNDO$\rangle$ in generation, allowing the model to refine the previous generated token on the fly. This effectively gives the model ability to explore reasoning paths, detect potential errors, and correct them within a single generation pass. To train the autoregressive sequence model in a non-monotonic manner, we first design a sequence augmentation method that inserts the true error tokens into the sequences, followed by $\langle$UNDO$\rangle$ tokens and correct references that teach models to detect and correct errors through $\langle$UNDO$\rangle$ tokens. With the error augmented sequences, directly training with fundamental supervised fine-tuning (SFT) may introduce negative learning (Kumar et al., 2025), where the model learn to generate error tokens. To this end, we design a masked SFT (mSFT) that masks out errors in the augmented sequences, enabling autoregressive sequence models to recognize and correct errors efficiently without introducing new errors. Afterward, we continue training the model via group relative policy optimization (GRPO) (Shao et al., 2024) with a multi-objective reward function to incentivize the model to correct revisions while penalizing stuttering, *i.e.*, unnecessary or incorrect erase behaviors. In this way, the trained model is capable of pruning and adjusting its generation trajectories to self-correct their outputs in a non-monotonic manner effectively and efficiently. We demonstrate the effectiveness of N-MARS through various benchmark datasets that require complex reasoning capabilities against autoregressive baselines. Moreover, we conduct comprehensive ablation studies to explore the design space of non-monotonic autoregressive sequence models, providing insights into key components such as the erase action design, sequence augmentation strategies, and training objectives. Our contributions are summarized as follows:

- **Non-Monotonic Autoregressive Approach.** We propose a novel sequence modeling paradigm that enables models to dynamically prune and adjust their generation trajectories on the fly to self-correct errors during inference.
- **Masked Supervised-Finetuning.** We introduce mSFT, which provides training signals for error identification, self-correction, and further error generation prevention. Moreover, we provide a theoretical analysis to demonstrate its effectiveness in mitigating error propagation.
- **Design Space Exploration.** We conduct a comprehensive design space exploration, offering actionable insights about non-monotonic autoregressive sequence models.
- **State-of-the-Art Performance.** We demonstrate that N-MARS achieves significant improvements over baseline methods on complex reasoning benchmarks, demonstrating the effectiveness of our proposed method.

**Conflict of Interest Disclosure.** Yiyue Qian, Yifan Ding, and Yan Liang are employed by Amazon. This work was conducted independently of their positions at Amazon and is unrelated to any Amazon product, service, or model. N-MARS is evaluated solely on publicly available models (Llama-3.2-1B, Llama-3.1-8B, and Qwen3-4B-Base) and public benchmarks; none of the evaluated models were developed by any author's employer. The authors declare no financial conflicts of interest.

## 2. Methodology

We formulate the autoregressive sequence modeling problem as Markov Decision Process (MDP), denoted as $\mathcal{M} = (\mathcal{S}, \mathcal{A}, \mathcal{P}, r, \gamma)$, with state space $\mathcal{S}$, finite action space $\mathcal{A}$, transition kernel $\mathcal{P} : \mathcal{S} \times \mathcal{A} \to \Delta(\mathcal{S})$, reward function $r : \mathcal{S} \times \mathcal{A} \to [R_{\min}, R_{\max}]$, and discount factor $\gamma \in (0, 1)$. The state space $\mathcal{S}$ is the set of all sequences, where each state $s_t = (x_1, \ldots, x_t)$ is a sequence of tokens from the vocabulary $\mathcal{V}$, and the action space $\mathcal{A}$ is a finite set of the vocabulary $\mathcal{V}$, *i.e.*, $\mathcal{A} = \mathcal{V}$. A policy $p_\theta$ parameterized by $\theta$ is a distribution over the next token on the current state:

$$p_\theta(x_{t+1}) = \prod_{i=1}^{T} p_\theta(x_{i+1}|x_{\leq i}). \tag{1}$$

In our case, the initial state is given by a special token $\langle$bos$\rangle$, and the end sequence token $\langle$eos$\rangle$ induces a terminal state. Moreover, the erase action is appended to the actions space, *i.e.*, $\mathcal{A} = \mathcal{V} \cup \{\langle$UNDO$\rangle\}$, where the action $\langle$UNDO$\rangle$ indicates to erase the last token in the current sequence.

### 2.1. Sequence Alignment Augmentation

To train a policy model $p_\theta$ that recognizes and corrects errors via the erase action, we introduce a data augmentation procedure based on sequence alignment (Myers, 1986). Given a prompt sequence $\mathbf{x}$ and reference completion $\mathbf{y} = (y_1, \ldots, y_n)$ from dataset $\mathcal{D}_{\text{ori}}$, we first sample a completion from the policy model:

$$\hat{\mathbf{y}} = (\hat{y}_1, \hat{y}_2, \ldots, \hat{y}_m) \sim p_\theta(\cdot|\mathbf{x}). \tag{2}$$

Afterward, we compute the maximal matching blocks between the reference and generated sequence:

$$\mathcal{M}(\mathbf{y}, \hat{\mathbf{y}}) = \{(a_i, b_i, \ell_i)\}_{i=1}^{K}, \text{ s.t. } y_{a_i:a_i+\ell_i} = \hat{y}_{b_i:b_i+\ell_i}, \tag{3}$$

where the decomposition is constrained to be:

- Maximal: No block can be extended in either direction.
- Non-overlapping: $a_i + \ell_i \leq a_{i+1}$ and $b_i + \ell_i \leq b_{i+1}$.
- Order-preserving: $a_1 < \cdots < a_K$ and $b_1 < \cdots < b_K$.

*Figure 1.* **The overall framework of N-MARS.** Left: N-MARS samples sequence in a non-monotonic manner. In a single forward pass, when a token ⟨UNDO⟩ is sampled as the next token, N-MARS erases the last token, and traces back to the previous token in the sequence. Right: The designed training paradigm of N-MARS. We first sample an initial response, then augment the sequence based on the sampled response by adding error tokens, followed by the same number of ⟨UNDO⟩ and the rest of the reference tokens. Afterward, we employ masked SFT that fine-tunes the model to identify and correct errors through token ⟨UNDO⟩ and further employ GRPO to incentivize correct behaviors while penalizing stuttering or under correction.

From this decomposition, we identify error indices as positions in $\hat{\mathbf{y}}$ not covered by any matching block:

$$\mathcal{E} = \{1, \ldots, m\} \setminus \bigcup_{i=1}^{K} [b_i, b_i + \ell_i). \quad (4)$$

The augmented training sequence $\mathbf{y}^*$ is constructed by processing $\hat{\mathbf{y}}$ sequentially, preserving matched tokens $\mathbf{p}$ while appending tokens ⟨UNDO⟩ and corrections $\mathbf{c}_i$ after each error span $\mathbf{e}_i \in \mathcal{E}$. For the segment between matching blocks $i$ and $i+1$:

$$\mathbf{y}^* = (\ldots, \underbrace{\hat{y}_{b_i}, \ldots, \hat{y}_{b_i+\ell_i-1}}_{\text{matched sequence } \mathbf{p}_i}, \underbrace{\hat{y}_{b_i+\ell_i}, \ldots, \hat{y}_{b_{i+1}-1}}_{\text{error span } \mathbf{e}_i},$$
$$\underbrace{\langle\text{UNDO}\rangle, \ldots, \langle\text{UNDO}\rangle}_{k_i \text{ erase actions } \mathbf{b}_i}, \underbrace{y_{a_i+\ell_i}, \ldots, y_{a_{i+1}-1}}_{\text{corrections } \mathbf{c}_i}, \ldots),$$
$$(5)$$

where $k_i = b_{i+1} - (b_i + \ell_i)$ denotes the number of error tokens between blocks $i$ and $i+1$. As a result, we obtain the augmented dataset $\mathcal{D}_{\text{aug}} = \{\{\mathbf{x}_j, \mathbf{y}_j^*\}\}_j$.

Besides the augmentation across real generations, we also explore two additional augmentation methods, *i.e.*, stochastic error injection and hard sample sequence augmentation, and discuss the comparisons and insights in Section 3.1.

## 2.2. Masked Supervised Fine-Tuning

To train policy model $p_\theta$ to understand the action ⟨UNDO⟩, we adopt supervised fine-tuning (SFT) on the error-correction traces in dataset $\mathcal{D}_{\text{aug}}$. However, standard SFT over these sequences induces negative learning (Kumar et al., 2025), since it explicitly minimizes the negative log-likelihood of error tokens, thereby encouraging error generation. To address this, we propose Masked Supervised Fine-Tuning (**mSFT**), a selective gradient optimization strategy. Formally, let $\mathbf{y}^* = (y_1, \ldots, y_T)$ be an augmented reference sequence from Equation 5, we define a binary mask $M \in \{0, 1\}^T$ where $M_t = 0$ if $y_t$ belongs to error spans, and $M_t = 1$ otherwise. Afterward, the mSFT objective is defined as:

$$\mathcal{L}_{\text{mSFT}}(\theta) = -\mathbb{E}_{(\mathbf{x}, \mathbf{y}^*) \sim \mathcal{D}_{\text{aug}}} \sum_{t=1}^{T} M_t \log p_\theta(y_t \mid y_{<t}, \mathbf{x}). \quad (6)$$

By masking the error positions via $M$, we effectively zero out the gradient contribution from error tokens, *i.e.*, $\nabla_\theta \mathcal{L}\,|_{y_t \in \mathbf{e}} = 0$. This ensures the model treats errors solely as context for detecting the need to erase, without being trained to generate them. Our proposed mSFT is theoretically grounded in Partial SGD (Mohtashami et al., 2022), which proves that stochastic gradient descent converges to

the optimal solution even when gradients are computed on a subset of dimensions, provided the masking preserves the target distribution. Mechanistically, this parallels the Selective Gradient Masking recently formalized for unlearning (Shilov et al., 2025), converting the error tokens into negative context (Liu et al., 2024) that conditions the subsequent repair without updating the policy towards error generation.

We now provide theoretical justification for mSFT. Our analysis establishes that masking error tokens during training yields improvements in error detection, corrective generation, and alleviates negative learning, comparing to SFT.

**Setup.** Let $(\mathbf{x}, \mathbf{y}^*) \sim \mathcal{D}_{\text{aug}}$ denote a backtrack trace where $\mathbf{y}^* = (\mathbf{p}, \mathbf{e}, \mathbf{b}, \mathbf{c})$ consists of a matched sequence $\mathbf{p}$, error span $\mathbf{e} = (e_1, \ldots, e_k)$, a sequence of erase tokens $\mathbf{b} = (\langle\text{UNDO}\rangle, \ldots, \langle\text{UNDO}\rangle)$ with $|\mathbf{b}| = k$, and correction $\mathbf{c}$, obtained in Section 2.1. The mask is $M_t = 0$ for positions $t \in \mathbf{e}$ and $M_t = 1$ otherwise. Both SFT and mSFT condition on the identical context for learning, and only differ in the tokens that contribute to the loss.

**Proposition 2.1** (mSFT Improves Self-Correction Learning). *Let $p_{\theta_{SFT}}$ and $p_{\theta_{mSFT}}$ be the policy models obtained by minimizing the SFT and Masked SFT objectives $\mathcal{L}_{SFT}$ and $\mathcal{L}_{mSFT}$ on augmented traces, respectively. Under smoothness, gradient conflict, and bounded gradient conditions (see Appendix B), a single gradient step satisfies:*

(a) *Error Detection: mSFT achieves lower expected $\langle\text{UNDO}\rangle$ cross-entropy:*

$$\mathbb{E}\left[\mathcal{L}_{\langle\text{UNDO}\rangle}(\theta^+_{mSFT})\right] \leq \mathbb{E}\left[\mathcal{L}_{\langle\text{UNDO}\rangle}(\theta^+_{SFT})\right] - \Gamma_{\langle\text{UNDO}\rangle}.$$
(7)

(b) *Corrective Generation: mSFT achieves lower expected correction cross-entropy:*

$$\mathbb{E}\left[\mathcal{L}_{\mathbf{c}}(\theta^+_{mSFT})\right] \leq \mathbb{E}\left[\mathcal{L}_{\mathbf{c}}(\theta^+_{SFT})\right] - \Gamma_{\mathbf{c}}.$$
(8)

(c) *Reduced Negative Learning: SFT increases error log-probability more than mSFT:*

$$\mathbb{E}\left[\log p_{\theta^+_{SFT}}(\mathbf{e} \mid \mathbf{x}, \mathbf{p})\right] \geq \mathbb{E}\left[\log p_{\theta^+_{mSFT}}(\mathbf{e} \mid \mathbf{x}, \mathbf{p})\right] + \Gamma_e.$$
(9)

*The gaps $\Gamma_{\langle\text{UNDO}\rangle}, \Gamma_{\mathbf{c}}, \Gamma_e$ are non-negative for sufficiently small learning rates $\eta$.*

*Proof Sketch.* Let $g_{\langle\text{UNDO}\rangle}$, $g_{\mathbf{c}}$, and $g_e$ denote the gradients of the $\langle\text{UNDO}\rangle$, correction, and error losses, respectively. SFT updates with the full gradient $g_{\langle\text{UNDO}\rangle} + g_e + g_{\mathbf{c}}$, while mSFT updates with the masked gradient $g_m = g_{\langle\text{UNDO}\rangle} + g_{\mathbf{c}}$.

Applying the Descent Lemma (Nesterov, 2013) to compare loss changes under both updates, the difference reduces to the inner product $\langle g_{\langle\text{UNDO}\rangle}, g_e \rangle$ (for part a) or $\langle g_{\mathbf{c}}, g_e \rangle$ (for part b). Under the gradient non-alignment assumption (Yu

et al., 2020), these inner products are non-positive in expectation, meaning the error gradient conflicts with learning $\langle\text{UNDO}\rangle$ and correction tokens. Removing $g_e$ from the update thus improves learning of both components. For part (c), since $\nabla_\theta \log p_\theta(\mathbf{e}) = -g_e$, the SFT update directly increases error probability via $\eta \|g_e\|^2$, while mSFT avoids this negative learning effect. Full proofs are in Appendix B. $\square$

**Discussion.** Proposition 2.1 provides theoretical justification for why masking error tokens during training improves self-correction learning. The results reveal a fundamental tension in training on augmented traces, *i.e.*, while the model observes error tokens to learn when to invoke $\langle\text{UNDO}\rangle$, optimizing their likelihood creates competing gradients that interfere with learning the correction mechanism itself. By masking error tokens, mSFT resolves this tension. Specifically, the model still conditions on errors, *i.e.*, preserving the context needed for self-correction, while avoiding reinforcing error generation. This yields simultaneous improvements in error detection (part a), corrective generation (part b), and reduced negative learning (part c).

### 2.3. Reinforcement Learning with Feedback

Following mSFT, we further employ reinforcement learning (RL) to refine the policy $p_\theta$, encouraging optimal utilization of $\langle\text{UNDO}\rangle$ and proactive error detection and correction.

**Reward Function.** The reward function $\mathcal{R}$ is computed after the raw trajectory $\tau \sim p_\theta(\cdot|\mathbf{x})$ for shaping the model behavior. To effectively guide the model's learning, we design a multi-objective reward function that (i) incentivizes final sequence correctness ($\mathcal{R}_{\text{inc}}$), while (ii) regulates the usage of the $\langle\text{UNDO}\rangle$ token based on outcome ($\mathcal{R}_{\text{pen}}$). The reward function is formulated as follows:

$$\mathcal{R}(\tau) = \mathcal{R}_{\text{inc}} + \mathcal{R}_{\text{pen}},$$
(10)

where $\mathcal{R}_{\text{inc}} \in \{-1, 1\}$ is a rule-based verifier that returns 1 if the final output $\tau^*$ is correct and $-1$ otherwise. Here $\tau^*$ denotes the post-processed sequence which we discuss in Section 2.4. The component $\mathcal{R}_{\text{pen}}$ regulates $\langle\text{UNDO}\rangle$ usage by rewarding erase actions when it leads to correct outputs and penalizing it when the output remains incorrect:

$$\mathcal{R}_{\text{pen}} = \begin{cases} +1.0, & \text{if } \mathcal{R}_{\text{inc}} = 1 \text{ and } N_{\langle\text{UNDO}\rangle}(\tau) \leq \kappa|\tau|, \\ +0.5, & \text{if } \mathcal{R}_{\text{inc}} = 1 \text{ and } N_{\langle\text{UNDO}\rangle}(\tau) > \kappa|\tau|, \\ -1.0, & \text{Otherwise,} \end{cases}$$
(11)

where $N_{\langle\text{UNDO}\rangle}(\tau)$ counts the number of $\langle\text{UNDO}\rangle$ tokens in the trajectory $\tau$, and $\kappa$ denotes a budget for $\langle\text{UNDO}\rangle$ tokens with respect to size $|\tau|$. When a correct output is obtained, the policy model is rewarded to the $\langle\text{UNDO}\rangle$ token usage, as it contributed to successful self-correction. Conversely,

*Table 1.* **N-MARS ablation experiments.** The highlighted row in gray is our proposed method.

*(a)* **Token Initialization.** Semantic-based initialization is most effective way for token initialization.

| Model | $\langle$UNDO$\rangle$ | GSM8K | MATH |
|---|---|---|---|
| Llama-3.2 (1B) | Cent. | 30.3 | 14.1 |
| | Ctx. | 31.2 | 14.9 |
| | Sem. | **31.3** | **15.2** |
| Qwen3 (4B) | Cent. | 84.0 | 68.4 |
| | Ctx. | 85.8 | 69.2 |
| | Sem. | **86.1** | **69.5** |

*(b)* **Supervised Training.** mSFT demonstrates best efficacy over $\mathcal{D}_{\text{aug}}$ that contains error tokens.

| Model | Method | GSM8K | MATH |
|---|---|---|---|
| Llama-3.2 (1B) | Base. | 8.8 | 3.2 |
| | SFT | 7.8 | 2.1 |
| | mSFT | **19.3** | **11.3** |
| Llama-3.1 (8B) | Base. | 53.3 | 23.1 |
| | SFT | 50.1 | 21.1 |
| | mSFT | **59.6** | **30.9** |

*(c)* **Reward Function.** Each component contributes to the overall performance, and our reward design outperforms prior arts.

| Model | $\mathcal{R}$ | GSM8K | MATH |
|---|---|---|---|
| Llama-3.2 (1B) | Both | **31.3** | **15.2** |
| | w/o $\mathcal{R}_{\text{inc}}$ | 28.3 | 14.1 |
| | w/o $\mathcal{R}_{\text{pen}}$ | 30.4 | 13.5 |
| | BSAFE+ | 31.3 | **15.2** |
| | SCoRe | 29.9 | 14.1 |
| | MGRPO | 30.2 | 14.8 |

*(d)* **GRPO.** Employing GRPO with our multi-objective rewards enhances model performance in reasoning tasks.

| Model | Method | GRPO | GSM8K |
|---|---|---|---|
| Llama-3.2 (1B) | Base. | ✗ | 8.8 |
| | | ✓ | 25.0 |
| | SFT | ✗ | 7.8 |
| | | ✓ | 23.9 |
| | mSFT | ✗ | 19.3 |
| | | ✓ | **31.3** |

*(e)* **Sequence Augmentation.** Learning erase actions from true error distributions brings more performance gain than others.

| Model | Aug. | GSM8K | MATH |
|---|---|---|---|
| Llama-3.2 (1B) | Rand. | 28.0 | 13.2 |
| | Hard | 30.9 | – |
| | Ours | **31.3** | **15.2** |
| Qwen3 (4B) | Rand. | 85.1 | 68.3 |
| | Hard | 85.8 | – |
| | Ours | **86.1** | **69.5** |

*(f)* **Positional Encoding.** Altering positional encodings does not contribute the result in N-MARS.

| Model | Position | GSM8K | MATH |
|---|---|---|---|
| Llama-3.2 (1B) | ✗ | **31.3** | **15.2** |
| | ✓ | 29.8 | 15.1 |
| Llama-3.1 (8B) | ✗ | **63.4** | **36.2** |
| | ✓ | 60.2 | 34.1 |
| Qwen3 (4B) | ✗ | **86.1** | **69.5** |
| | ✓ | 81.4 | 65.6 |

when the output is incorrect, the policy model is penalized, as it represents wasted computation that failed to improve the result. Furthermore, this reward design **encourages early error detection**, *i.e.*, since correcting an error at position $t$ requires erasing all tokens generated after $t$, early detection yields a more favorable reward controlled by $\kappa$.

**Optimization.** Given the reward function $\mathcal{R}(\tau, \mathbf{y}^*)$, we optimize the policy model $p_\theta$ using Group Relative Policy Optimization (GRPO) (Shao et al., 2024). For each prompt $\mathbf{x}$, we sample a group of $G$ trajectories $\{\tau_i\}_{i=1}^G$ from the current policy, compute rewards $r_i = \mathcal{R}(\tau_i, \mathbf{y}^*)$, and derive group-normalized advantages:

$$A_i = \frac{r_i - \text{mean}(\{r_j\}_{j=1}^G)}{\text{std}(\{r_j\}_{j=1}^G)}. \tag{12}$$

Afterward, the GRPO objective is defined as:

$$\mathcal{L}_{\text{GRPO}}(\theta) = -\mathbb{E}_{\{\mathbf{x}, \mathbf{y}^*\} \sim \mathcal{D}_{\text{aug}}, \{\tau_i\}_{i=1}^G \sim p_{\theta_{\text{old}}}(\cdot|\mathbf{x})} \tag{13}$$

$$\left[ \frac{1}{G} \sum_{i=1}^G \min\left( \rho_i A_i, \text{clip}(\rho_i, 1-\epsilon, 1+\epsilon) A_i \right) \right] \tag{14}$$

$$+ \beta D_{\text{KL}}(p_\theta \| p_{\text{ref}}), \tag{15}$$

where $\rho_i = p_\theta(\tau_i|\mathbf{x})/p_{\theta_{\text{old}}}(\tau_i|\mathbf{x})$ is the importance ratio, $\epsilon$ is the clipping threshold, $\beta$ is the KL penalty coefficient, and $p_{\text{ref}}$ is the reference policy. Through GRPO with our multi-objective reward $\mathcal{R}$, the model is jointly trained to produce accurate final outputs and to utilize the $\langle$UNDO$\rangle$ token judiciously, *i.e.*, erasing only when it yields demonstrable performance gains.

### 2.4. Inference as Non-Monotonic Generation

We view the inference process as non-monotonic generation, where the semantic content of the sequence is not strictly append-only. Specifically, the model generates a raw trajectory $\tau \sim p_\theta(\cdot|\mathbf{x})$ autoregressively, which is then reduced to a final sequence $\mathbf{y}$ through a deterministic stack-based operator, as illustrated in Figure 1. Standard tokens are pushed onto the stack, while the $\langle$UNDO$\rangle$ token triggers a pop operation that effectively deletes the previous token in the stack. This decoupling of the generation from the semantic sequence allows the model to explore and prune its own reasoning space dynamically, leading to non-monotonic generation. For example, the trajectory $\tau = (\text{The}, \text{answer}, \text{is}, 5, \langle\text{UNDO}\rangle, 4)$ is converted to the final sequence $\mathbf{y} = (\text{The}, \text{answer}, \text{is}, 4)$, seamlessly correcting the error.

## 3. Design Spaces and Insights

We conduct a comprehensive ablation study to provide more insights about our framework design, including token initialization, supervised training, reward function design, reinforcement learning training, sequence augmentation, and positional encoding. The result is listed in Table 1.

### 3.1. Training Recipe

**Token Initialization.** We explore three token initialization methods for $\langle$UNDO$\rangle$ token, *i.e.*, centroid (Cent.), context-based (Ctx.), and semantic anchor (Sem.) initializations. The detailed discussions about these methods are provided in Appendix C. According to Table 1a, both context and

semantic-based strategies outperform the naive centroid method, with semantic initialization achieving the highest accuracy on both GSM8K and MATH-500 datasets. Notably, the performance gap between informed initialization methods, *i.e.*, context and semantic, and the centroid method is more substantial on the challenging MATH-500 benchmark, indicating that *semantically meaningful token embeddings provide greater benefits for complex reasoning tasks.*

**Supervised Fine-Tuning.** We compare mSFT against standard SFT using the augmented dataset $\mathcal{D}_{aug}$. As shown in Table 1b, standard SFT exhibits negative learning, evidenced by a performance decline compared to baseline (Few-Shot). This degradation likely stems from error tokens within $\mathcal{D}_{aug}$ leading the trained models to imitate error tokens and generate erroneous outputs. Conversely, mSFT mitigates this by masking error tokens, thereby yielding a performance gain.

**Reward Function.** We examine reward function designs in Table 1c. Both components contribute distinctly, *i.e.*, $\mathcal{R}_{inc}$ provides a 10% gain on GSM8K, whereas $\mathcal{R}_{pen}$ yields a 12% gain on MATH-500. This suggests that *simpler tasks (GSM8K) benefit primarily from outcome-based rewards*, where correct erasing naturally emerges from accuracy signals. Conversely, *complex reasoning tasks require explicit regularization of erase behavior to prevent degenerate strategies*, *i.e.*, either over-reliance or under-utilization of the undo mechanism.

Besides our proposed reward functions, we also test the designs in BSAFE+ (Sel et al., 2025), SCoRe (Kumar et al., 2025), and MGRPO (Ding et al., 2025). Implementation details are provided in Appendix D. We find that our designed reward function performs better than those in SCoRe and MGRPO. Although BSAFE+ achieves comparable performance with our proposed method, it employs an external LLM as a critic to compute the reward, which introduces computational overhead. Therefore, our proposed method remains a better design in terms of performance and computational costs.

**GRPO.** We assess the efficacy of GRPO over base and fine-tuned models in Table 1d. Note that the SFT result is the GRPO training based on the model after SFT over $\mathcal{D}_{aug}$. GRPO consistently enhances performance across different settings, even for the SFT model that has shown negative learning in Table 1b. This indicates that *GRPO effectively contributes to N-MARS's generate steps, leading to improved reasoning capabilities.*

### 3.2. Augmentation

**Sequence Augmentation.** We examine the proposed sequence augmentation method in Section 2.1, along with two additional approaches for erase, *i.e.*, stochastic error injection (Rand.) and hard sample sequence augmentation (Hard). Appendix E provides the details of these two augmentation methods. Table 1e demonstrates that our proposed method outperforms both alternatives, indicating that *targeted error injection focusing on critical reasoning steps is more effective than random or hard sample-based augmentations for training robust non-monotonic models.*

**Positional Encoding.** As discussed in Cundy & Ermon (2024), altering the positional encoding may also bring potential performance gains. The intuition is that correction tokens should inherit the position indices of the error tokens they replace, such that the final semantic sequence appears positionally contiguous. We investigate this approach in Table 1f, and we find that positional encodings consistently degrade performance. We attribute this to RoPE (Su et al., 2024), which rewinds position indices, introducing non-monotonic discontinuities that conflict with pretrained attention patterns. Moreover, *standard monotonic positions already preserve the full context of the error-correction process*, allowing the model to learn correction semantics directly from the ⟨UNDO⟩ token and its surrounding context rather than relying on positional cues.

## 4. Experiment

We evaluate the proposed method N-MARS on three benchmarks, *i.e.*, GSM8K (Cobbe et al., 2021), MATH-500 (Hendrycks et al., 2021), and MBPP (Austin et al., 2021). We employ autoregressive models, *i.e.*, Llama-3.2-1B (Meta, 2024), Llama-3.1-8B (Grattafiori et al., 2024), and Qwen3-4B-Base (Yang et al., 2025), as backbone models. Implementation details are provided in Appendix A.

### 4.1. Main Result

Table 2 lists the performance comparison of N-MARS against baseline methods on three benchmark datasets. N-MARS achieves SOTA performance across all three backbone LLMs with relative improvements of 11.4% and 13.4% over the strongest baseline method, *i.e.*, ReVISE, on GSM8K and MATH, respectively, with backbone Llama-3.2-1B. This suggests that *non-monotonic generation particularly benefits on capability-constrained models*, where the ability to erase and correct errors compensates for limited reasoning capacity. Moreover, the performance gap between N-MARS and baselines widens on MATH compared to GSM8K, indicating that *complex reasoning tasks stand to gain more from non-monotonic generation*, as the likelihood of errors increases with task complexity. Beyond mathematical reasoning, N-MARS also improves coding performance on MBPP, demonstrating that *the benefits of non-monotonic generation extend to diverse reasoning domains beyond*

*Table 2.* **Main performance comparison.** The best performance is highlighted in bold, and the runner-up is underlined.

| Method | Llama-3.2-1B | | | Llama-3.1-8B | | | Qwen3-4B | | |
|---|---|---|---|---|---|---|---|---|---|
| | GSM8K | MATH-500 | MBPP | GSM8K | MATH-500 | MBPP | GSM8K | MATH-500 | MBPP |
| Few-Shot | 8.8 | 3.2 | 25.1 | 53.3 | 24.1 | 48.0 | 72.3 | 54.1 | 67.6 |
| SFT | 14.3 | 10.5 | 29.9 | 55.6 | 28.1 | 58.8 | 81.5 | 60.7 | 70.2 |
| RFT | 26.1 | 12.7 | 29.5 | 59.1 | 30.9 | 58.6 | 80.3 | 63.8 | 71.3 |
| STaR+ | 26.5 | 12.9 | 30.7 | 59.3 | 30.6 | 59.1 | 82.1 | 64.0 | 70.9 |
| ReVISE | 28.1 | 13.4 | 33.3 | 61.6 | 33.6 | 60.3 | 84.2 | 66.3 | 71.9 |
| **Ours** | **31.3** | **15.2** | **34.1** | **63.4** | **36.2** | **61.1** | **86.1** | **69.5** | **72.5** |

*Table 3.* **Domain Transfer.** GSM → MATH indicates training on GSM8K and testing on MATH-500, vice versa.

| Model | Method | GSM → MATH | MATH → GSM |
|---|---|---|---|
| Llama-3.2 (1B) | SFT | 2.6 | 7.3 |
| | ReVISE | 10.2 | 8.8 |
| | **Ours** | **11.4** | **9.2** |
| Llama-3.1 (8B) | SFT | 22.8 | 54.3 |
| | ReVISE | 27.0 | 58.4 |
| | **Ours** | **28.5** | **60.1** |

*Table 4.* **Comparison with instruction-tuned model.**

| Model | Method | GSM8K | MATH | MBPP |
|---|---|---|---|---|
| Llama-3.2 Instruct (1B) | Few-shot | 44.4 | 30.6 | 43.8 |
| | SFT | 41.9 | 26.9 | 40.4 |
| | ReVISE | 52.3 | 31.5 | 54.2 |
| | **Ours** | **53.1** | **33.0** | **55.6** |

*arithmetic reasoning.*

### 4.2. Additional Analysis

**Generalization under Domain Shift.** We evaluate the generalization ability of N-MARS under dataset domain shift settings and report the results in Table 3. N-MARS consistently outperforms all baseline methods across different model scales and transfer directions. Specifically, N-MARS achieves an 11.8% relative improvement over ReVISE under the GSM8K → MATH setting and a 4.5% gain under MATH → GSM8K, demonstrating strong generalization ability and effective domain transferability of our proposed method.

**Performance over Instruction-tuned Models.** We further evaluate N-MARS on instruction-tuned autoregressive sequence models and report the results in Table 4. We observe that: (i) SFT over the instruction-tuned models underperforms few-shot prompting, likely due to the catastrophic forgetting (Kirkpatrick et al., 2017; Luo et al., 2025), hindering models from learning new information effectively with

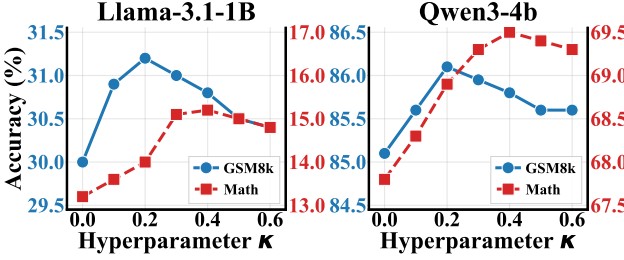

*Figure 2.* **Hyperparameter analysis for** $\kappa$. The hyperparameter $\kappa$ is chosen from $\{0, 0.1, 0.2, 0.3, 0.4, 0.5, 0.6\}$.

a small-sized dataset. (ii) Our proposed method N-MARS consistently outperforms all baselines, demonstrating its effectiveness on instruction-tuned backbone models.

**Hyperparameter Analysis.** The token ⟨UNDO⟩ budget $\kappa$ serves as a critical hyperparameter controlling the trade-off between correction capability and generation efficiency. We evaluate the impact of varying $\kappa \in \{0, 0.1, 0.2, 0.3, 0.4, 0.5, 0.6\}$ across two distinct models, *i.e.*, Llama-3.2-1B and Qwen3-4B, on both GSM8K and MATH datasets in Figure 2. We observe that the optimal $\kappa$ is highly correlated with task complexity. For the simpler GSM8K benchmark, performance peaks at a lower budget of $\kappa = 0.2$. Beyond this point, performance slightly degrades, suggesting that excessive budgets on simpler tasks may encourage "over-deliberation", *i.e.*, unnecessary edits that introduce generation noise without improving accuracy. Conversely, for the more challenging MATH dataset, the performance curve shifts rightward, peaking at $\kappa = 0.4$. This indicates that complex reasoning chains require a larger correction window to effectively rectify deep errors. Notably, this pattern is consistent across model scales. Both the 1B and 4B models exhibit the same optimal $\kappa$ values for respective datasets, suggesting that the correction budget is a property of the task's reasoning depth rather than the model capacity. Based on these empirical findings, we adopt a task-adaptive configuration, setting $\kappa = 0.2$ for GSM8K and $\kappa = 0.4$ for MATH in our main experiments to maximize

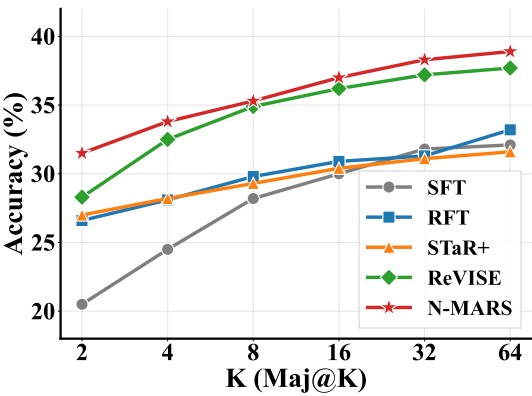

*Figure 3.* **Test-time Scaling Analysis.** Majority voting accuracy (maj@$k$) on GSM8K with Llama-3.2-1B as the number of samples $k$ increases from 2 to 64.

*Table 5.* **Generation budget scaling and efficiency** under generation budgets (tokens). The average number of newly generated tokens and the wall-clock latency are measured at the 256 budget.

| Dataset | Method | Acc @ Budget | | | Avg. New Tokens | Latency (mins) |
|---|---|---|---|---|---|---|
| | | 128 | 256 | 512 | | |
| GSM8K | Base | 5.1 | 8.8 | 8.8 | 118.3 | 7.2 |
| | ReVISE | 27.2 | 28.1 | 28.1 | 178.9 | 9.8 |
| | **Ours** | **29.6** | **31.3** | **31.3** | 153.7 | 9.1 |
| MATH | Base | 2.4 | 3.2 | 3.2 | 174.4 | 9.4 |
| | ReVISE | 12.0 | 13.4 | 13.4 | 249.3 | 11.7 |
| | **Ours** | **13.1** | **15.2** | **15.9** | 217.5 | 10.4 |

the efficiency-accuracy trade-off.

**Test-time Scaling.** We evaluate test-time scaling by measuring majority voting accuracy (maj@$k$) as the number of sampled responses increases from 2 to 64 (Figure 3). SFT-based methods (SFT, RFT, STaR+) exhibit diminishing returns: RFT improves only $+6.6\%$ and STaR+ only $+4.6\%$ from maj@2 to maj@64, suggesting that these methods converge to a narrow distribution with limited diversity. In contrast, RL-based methods demonstrate stronger scaling: ReVISE gains $+9.4\%$ across the same range. N-MARS achieves the best scaling behavior, starting from a higher baseline and maintaining consistent gains to reach $38.9\%$ at maj@64. This advantage stems from non-monotonic generation, *i.e.*, the ability to backtrack and explore alternative reasoning paths, which yields a more diverse sample distribution, which majority voting effectively exploits.

**Generation Budget Scaling and Efficiency.** We study the performance and cost scale with the generation budget by varying it over $\{128, 256, 512\}$ tokens, reporting accuracy alongside the average number of newly generated tokens and wall-clock latency at the 256 budget in Table 5. N-MARS dominates at every budget on both datasets, improving by $11.4\%$ over ReVISE on GSM8K at the 256 budget and by

*Table 6.* **Comparison with self-correction methods**, including token-level (Self-Reflect) and step-level (Self-Backtrack) methods.

| Model | Method | GSM8K | MATH-500 |
|---|---|---|---|
| Llama-3.2 (1B) | Self-Reflect | 25.8 | 11.6 |
| | Self-Backtrack | 26.4 | 11.2 |
| | **Ours** | **31.3** | **15.2** |
| Llama-3.1 (8B) | Self-Reflect | 60.9 | 34.5 |
| | Self-Backtrack | 60.9 | 31.7 |
| | **Ours** | **63.4** | **36.2** |

$18.7\%$ on MATH-500 at the 512 budget. We further observe that the budget at which accuracy saturates tracks task complexity. On the simpler GSM8K, N-MARS already peaks at the 256 budget, whereas on the harder MATH-500 it keeps improving up to the 512 budget, consistent with complex reasoning chains requiring a larger correction window to rectify deep errors. Moreover, this higher accuracy does not come at the expense of efficiency. At the 256 budget, N-MARS attains the best accuracy while generating *fewer* tokens than ReVISE and incurring lower latency. This indicates that N-MARS corrects errors in place rather than re-deriving long reasoning chains, demonstrating a strictly better accuracy-efficiency trade-off than baseline methods.

**Comparison with Self-Correction Methods.** We further compare N-MARS against two alternative correction paradigms in Table 6, *i.e.*, a verbal self-reflection method (Self-Reflect) (Madaan et al., 2023; Shinn et al., 2023), trained on the same augmented traces but using natural-language revision markers instead of the ⟨UNDO⟩ token, and the step-level Self-Backtrack (Yang et al., 2026). N-MARS consistently outperforms both across backbones and datasets, with relative improvements of up to $35.7\%$ over Self-Backtrack and $31.0\%$ over Self-Reflect on MATH-500 with Llama-3.2-1B. Two distinctions explain this advantage. First, unlike verbal reflection, which must additionally learn the precise error starting position to remove erroneous tokens (otherwise they linger in the final output), the explicit ⟨UNDO⟩ token yields a *clean* corrected sequence by construction, confirming the necessity of an explicit erase mechanism. Second, Self-Backtrack operates at the reasoning-step level and thus requires explicit step boundaries, whereas N-MARS operates at the token level, *i.e.*, the atomic unit of autoregressive generation, making it task-agnostic and naturally extensible beyond structured reasoning to domains such as code generation and open-ended text.

## 5. Related Work

Autoregressive models, exemplified by large language models, generate sequences token-by-token in a monotonic manner, *i.e.*, once a token is emitted, later tokens must

condition on it, and the model cannot revise earlier decisions (Bachmann & Nagarajan, 2024; Thoppilan et al., 2022). A fundamental limitation of this paradigm is error propagation (Zhang et al., 2024a; Gan et al., 2025), where a mistake made early in generation can propagate and amplify over the remainder of the sequence, since there is no mechanism for erasing flawed intermediate content. Recent studies further show that such systems remain highly sensitive to noisy, adversarial, or misleading prompts in interactive settings, often causing harmful or degraded downstream behaviors (Chen et al., 2025a;b).

### 5.1. Post-hoc Verification and Iterative Refinement

A common direction to tackle the error propagation issue is post-hoc self-critique and revision (Shinn et al., 2023; Welleck et al., 2023; Gou et al., 2024; Wang et al., 2023). These methods typically follow an iterative loop, *i.e.*, generate an initial draft, produce feedback through the same model or an auxiliary model, and then regenerate a revised answer conditioned on the critique. For instance, Self-Refine (Madaan et al., 2023) demonstrates that models can improve responses via multiple rounds of self-generated feedback, without task-specific fine-tuning or additional supervision: the same pretrained model alternates between drafting, critiquing, and revising. Reflexion (Shinn et al., 2023) extends this idea to agentic settings by maintaining an explicit memory of past failures. The agent reflects on these errors and uses them as "lessons" to guide future attempts. More recently, intrinsic self-verification aims to endow the model with an internal judgment of whether its partial reasoning or output is trustworthy, reducing reliance on external critics. ReVISE (Lee et al., 2025) trains a model to decide whether a reasoning trajectory is sound or should be revised. While effective, post-hoc refinement and self-verification pipelines typically introduce additional inference overhead because they require at least one extra generation pass to critique and revise.

### 5.2. Self-Refine via Backtrack Generation

The most relevant line of work to our proposed method is self-refinement via backtrack generation, which allows the model to interleave generation with on-the-fly self-assessment and revision within a single decoding pass. These works mainly focus on safety violations in autoregressive sequence models. For example, Zhang et al. (2025a) proposes a backtracking-enabled decoding mechanism for safer generation by introducing a special [RESET] token and fine-tuning the model to emit it when the output drifts toward unsafe content. Sel et al. (2025) extends this idea by introducing category tokens [CATEGORY] that flag the type of safety violation, enabling targeted revisions. Wu et al. (2025) introduces REVERSE for vision-language generation, training a model to interleave generation with self-

refinement by inserting special tokens, *i.e.*, <UN> and <CN>, that trigger verification and correction behaviors. Unlike prior backtracking methods that focus primarily on safety violations (Zhang et al., 2025a; Sel et al., 2025) or vision-language tasks (Wu et al., 2025), N-MARS targets complex reasoning domains where error propagation is particularly detrimental. Moreover, N-MARS employs a fine-grained non-monotonic generation strategy that allows the model to revise arbitrary segments of the generated sequence, rather than only from fixed positions, *i.e.*, end of sentences or end of generation. This flexibility is crucial for reasoning tasks, where errors can occur at any step, and earlier mistakes may require revisiting multiple subsequent tokens.

## 6. Conclusion

In this work, we present N-MARS, a novel non-monotonic autoregressive sequence model that empowers models with the ability to self-correct errors during generation by dynamically pruning and adjusting their generation trajectories on the fly. To train N-MARS efficiently, we propose a training paradigm that contains (i) sequence augmentation that provides training signal from true error distribution for the correction mechanism, (ii) masked SFT which selectively masks error tokens during training to prevent negative learning while teaching the model to detect and correct errors, (iii) GRPO with a multi-objective reward function that incentivizes correct final outputs while penalizing unnecessary or incorrect erase behaviors, encouraging early error detection and efficient self-correction. Moreover, we conduct comprehensive experiments to explore the design space of N-MARS and provide actionable insights into model architecture and training design. We further compare N-MARS with existing SFT- and RL-based baseline methods over benchmark datasets, demonstrating the effectiveness of our proposed method.

### Acknowledgements

This work is partially supported by the NSF under grants IIS-2533550, IIS-2321504, IIS-2217239, CNS-2426514, and CMMI-2146076, Notre Dame Strategic Framework Research Grant (2025), and Notre Dame Poverty Research Package (2025). This work leverage Delta at NCSA through allocation CIS260040 from the ACCESS program (Boerner et al., 2023), which is supported by U.S. National Science Foundation grants #2138259, #2138286, #2138307, #2137603, and #2138296. Any expressed opinions, findings, conclusions, and recommendations are those of the authors and do not necessarily reflect the views of the sponsors.

## Impact Statement

This work introduces N-MARS, a non-monotonic autoregressive framework that enables sequence models to self-correct errors during generation. We discuss the broader implications of this research below.

**Potential Benefits.** The ability to detect and correct errors on-the-fly addresses a fundamental limitation of current autoregressive models, *i.e.*, error propagation. By enabling models to revise their outputs within a single generation pass, N-MARS reduces the computational overhead associated with external verification systems, post-hoc correction pipelines, and multi-pass generation strategies. This efficiency gain is particularly valuable for resource-constrained deployment scenarios and real-time applications where latency is critical. Furthermore, the self-correction mechanism may improve the reliability of model-generated content in high-stakes domains such as medical information retrieval, legal document drafting, and educational content generation, where factual accuracy is paramount. More broadly, the non-monotonic generation paradigm opens new avenues for research into more interactive and adaptive sequence models that can iteratively refine their outputs based on internal feedback, potentially leading to more robust and trustworthy AI systems.

**Risk and Misuse Considerations.** As with many advances in sequence models, improved self-correction capabilities may lower barriers to generating coherent and persuasive text. In adversarial settings, such capabilities could be misused to produce more convincing misinformation or to refine deceptive content that evades detection. While N-MARS itself does not introduce new knowledge sources or factual grounding mechanisms beyond existing autoregressive frameworks, its improved error recovery could amplify the fluency and apparent reliability of downstream applications. We therefore emphasize that responsible deployment should follow existing best practices in content moderation, fact-checking integration, and output verification, and that access to models with enhanced self-correction capabilities should be carefully managed in sensitive domains.

**Limitations and Scope.** Our approach has several limitations that bound its current applicability. First, in this work, we employ LLMs as autoregressive sequence models since they are current state-of-the-art, and the effectiveness of N-MARS may vary with different model architectures or training paradigms. Second, the current framework operates at the token level; extending to higher-level semantic units, *i.e.*, phrases, sentences, or reasoning steps, is an open challenge. Finally, the computational overhead of the non-monotonic autoregressive model, while lower than external verification methods, still introduces additional generation steps that may accumulate in complex reasoning chains.

**Future Directions.** This work opens several avenues for future research. Current work focuses on post-training to introduce self-correction; integrating non-monotonic generation capabilities during initial model pretraining could yield more robust correction behaviors. Exploring hierarchical backtracking mechanisms—where models can revise at multiple granularities, *i.e.*, token, phrase, paragraph, may further improve correction efficiency. The masked SFT framework could be extended to other learning-from-demonstration settings where training data contains systematic errors. Finally, investigating the interplay between self-correction and reasoning chain coherence in long-horizon tasks presents an important direction for deploying N-MARS in complex, multi-step problem-solving scenarios.

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

# A. Implementation Details

### A.1. Environment

All experiments are conducted on Linux OS equipped with four Nvidia A100 GPUs. The models are implemented using PyTorch 2.4.0 with CUDA 12.1 and Python 3.11.5.

### A.2. Benchmark Datasets

- **GSM8K.** GSM8K (Cobbe et al., 2021) is a dataset of 8,792 high-quality linguistically diverse grade school math word problems. The dataset was created to support question-answering for basic mathematical problems that require multi-step reasoning. We use the default train-test splits to ensure fairness and consistency across all experiments.
- **MATH-500.** This benchmark (Hendrycks et al., 2021) consists of challenging problems from high school mathematics competitions, *e.g.* AMC 10, AMC 12, and AIME, etc. Each problem in MATH has a full, step-by-step solution that can be used to teach models for generating answer derivations and explanations.
- **MBPP.** MBPP (Austin et al., 2021) is a benchmark that contains 974 crowd-sourced Python programming problems, designed to be solvable by entry-level programmers, covering programming fundamentals and standard library functionality. Each problem consists of a task description, code solution, and 3 automated test cases. We follow ReVISE (Lee et al., 2025) to obtain the training samples. Specifically, we apply CoT to GPT-4o to obtain 16 valid reasoning paths, along with corresponding code solutions that pass all test cases, as the training set.

### A.3. Backbone Autoregressive Sequence Models

We choose the state-of-the-art open-source LLMs, Llama-3.2-1B, Llama-3.1-8B, and Qwen3-4B, as our backbone autoregressive sequence models. We obtain these model checkpoints from official releases in the Huggingface Library by Meta AI and Qwen AI, respectively.

### A.4. Baseline Methods

We compare N-MARS against the following baseline methods.

- **SFT** (Supervised Fine-Tuning). The model is fine-tuned on ground-truth reasoning traces from the training set. This serves as the basic baseline that learns from expert demonstrations without any self-generated data.
- **RFT** (Yuan et al., 2023). The model first generates multiple candidate solutions per problem, then filters to retain only correct solutions (verified by answer matching). The model is subsequently fine-tuned on this self-generated, correctness-filtered dataset.
- **STaR+** (Zelikman et al., 2022). An iterative extension of RFT where the generate-filter-train loop is repeated for multiple rounds. In each iteration, the model generates solutions, the correct ones are kept, and the model is fine-tuned on the accumulated dataset.
- **ReVISE** (Lee et al., 2025). A two-stage approach that first trains a model via SFT on correct reasoning traces, then applies reinforcement learning to learn revision behavior. At inference time, the model can generate an initial solution and subsequently revise it in a second pass. Unlike our approach, ReVISE requires multiple generation passes for revision.

### A.5. Training Details

We use AdamW optimizer with learning rate $\eta = \{10^{-4}, 10^{-5}\}$ with 0.1 warmup ratio and 0.1 weight decay for all experiments. Training epochs range from 1 to 3. For supervised fine-tuning, the training batch size is set to 16 and 32 for evaluation. For reinforcement learning, the batch size is set to 4 for training with number of generation $G = 8$. The $\langle \text{UNDO} \rangle$ budget $\kappa$ is set to $\kappa = 0.2$ for GSM8K and $\kappa = 0.4$ for MATH and MBPP.

### A.6. Evaluation Setup

We utilize lm-evaluation-harness (Gao et al., 2024) for evaluation to ensure consistency across all experiments. Specifically, for all benchmark datasets, *i.e.*, GSM8K, MATH and MBPP, we use the test splits provided on huggingface datasets library (Lhoest et al., 2021).

# B. Proofs

In this section, we provide detailed proofs for Proposition 2.1 (mSFT Improves Self-Correction Learning) presented in the main paper. We first restate the proposition and assumptions for completeness, then prove each part.

## B.1. Setup and Notation

Let $(\mathbf{x}, \mathbf{y}^*) \sim \mathcal{D}_{\text{aug}}$ be an augmented trace where $\mathbf{y}^* = (\mathbf{p}, \mathbf{e}, \mathbf{b}, \mathbf{c})$ consists of a matched prefix $\mathbf{p}$, an error span $\mathbf{e} = (e_1, \ldots, e_k)$, erase tokens $\mathbf{b} = (\langle\text{UNDO}\rangle, \ldots, \langle\text{UNDO}\rangle)$, with size $|\mathbf{b}| = k$, and correction tokens $\mathbf{c}$. We use $\eta > 0$ to denote the learning rate, and let $p_\theta(\cdot \mid \mathbf{x})$ be the model's conditional distribution parameterized by $\theta$.

The mask $M$ is defined as $M_t = 0$ for positions $t \in \mathbf{e}$ (error tokens) and $M_t = 1$ otherwise. The training objectives are:

$$\mathcal{L}_{\text{SFT}}(\theta) = -\mathbb{E}_{(\mathbf{x}, \mathbf{y}^*) \sim \mathcal{D}_{\text{aug}}} \left[ \sum_{t=1}^{T} \log p_\theta(y_t^* \mid y_{<t}^*, \mathbf{x}) \right], \tag{16}$$

$$\mathcal{L}_{\text{mSFT}}(\theta) = -\mathbb{E}_{(\mathbf{x}, \mathbf{y}^*) \sim \mathcal{D}_{\text{aug}}} \left[ \sum_{t=1}^{T} M_t \cdot \log p_\theta(y_t^* \mid y_{<t}^*, \mathbf{x}) \right]. \tag{17}$$

**Assumption B.1** (Smoothness). Along the optimization trajectory $\{\theta_t\}_{t=0}^{T}$, the loss functions $\mathcal{L}_{\langle\text{UNDO}\rangle}(\theta) := -\log p_\theta(\langle\text{UNDO}\rangle | \mathbf{x}, \mathbf{p}, \mathbf{e})$, correction loss $\mathcal{L}_{\mathbf{c}}(\theta)$, and error-score $s_\theta := \log p_\theta(\mathbf{e} | \mathbf{x}, \mathbf{p})$ are $L$-smooth, i.e., $\|\nabla_\theta f(\theta_1) - \nabla_\theta f(\theta_2)\| \leq L\|\theta_1 - \theta_2\|$ for all iterates $\theta_1, \theta_2$ on the trajectory.

**Assumption B.2** (Gradient Non-Alignment). As discussed in Yu et al. (2020), we assume the gradients are non-positively aligned in expectation. Specifically, let $g_{\langle\text{UNDO}\rangle} = \nabla_\theta \mathcal{L}_{\langle\text{UNDO}\rangle}(\theta)$, $g_{\mathbf{c}} = \nabla_\theta \mathcal{L}_{\mathbf{c}}(\theta)$, and $g_e = \nabla_\theta \mathcal{L}_e(\theta)$ denote the true population gradients. We assume:

(a) $\langle\text{UNDO}\rangle$**-Error Non-Alignment:** There exists $\alpha_{\langle\text{UNDO}\rangle} \geq 0$ such that

$$\mathbb{E}\left[ -\langle g_{\langle\text{UNDO}\rangle}, g_e \rangle \right] \geq \alpha_{\langle\text{UNDO}\rangle} \cdot \mathbb{E}\left[ \|g_e\|^2 \right]. \tag{18}$$

(b) **Correction-Error Non-Alignment:** There exists $\alpha_{\mathbf{c}} \geq 0$ such that

$$\mathbb{E}\left[ -\langle g_{\mathbf{c}}, g_e \rangle \right] \geq \alpha_{\mathbf{c}} \cdot \mathbb{E}\left[ \|g_e\|^2 \right]. \tag{19}$$

When $\alpha > 0$, the gradients are in conflict; when $\alpha = 0$, they are orthogonal or non-negatively aligned on average.

**Assumption B.3** (Bounded Gradient Ratios). There exists a constant $C \geq 1$ such that for $g_m = g_{\langle\text{UNDO}\rangle} + g_{\mathbf{c}}$:

$$\mathbb{E}\left[ \|g_m\|^2 + \|g_m + g_e\|^2 \right] \leq C \cdot \mathbb{E}\left[ \|g_e\|^2 \right]. \tag{20}$$

This holds when error gradients are not negligible relative to the other gradient components.

**Remark.** Assumption B.3 is satisfied by construction in our framework: the augmented dataset $\mathcal{D}_{\text{aug}}$ is designed such that every training sequence contains error tokens followed by correction tokens (see Section 2.1). This guarantees that $g_e \neq 0$ for all training samples, ensuring the error gradient norm $\mathbb{E}[\|g_e\|^2]$ remains non-negligible throughout training. Consequently, the bounded ratio condition holds with a finite constant $C$ that depends on the relative magnitudes of the $\langle\text{UNDO}\rangle$, correction, and error token gradients within each augmented trace.

**Lemma B.4** (Descent Lemma (Nesterov, 2013)). *Let $f : \mathbb{R}^d \to \mathbb{R}$ be an $L$-smooth function, i.e., $\|\nabla f(\theta_1) - \nabla f(\theta_2)\| \leq L\|\theta_1 - \theta_2\|$ for all $\theta_1, \theta_2$. Then for any $\theta$ and displacement $u \in \mathbb{R}^d$:*

$$|f(\theta + u) - f(\theta) - \langle \nabla f(\theta), u \rangle| \leq \frac{L}{2} \|u\|^2. \tag{21}$$

*Equivalently, there exists a remainder $r$ with $|r| \leq \frac{L}{2}\|u\|^2$ such that $f(\theta + u) = f(\theta) + \langle \nabla f(\theta), u \rangle + r$.*

## B.2. Proof of Proposition 2.1 Part (a)

**Proposition B.5** (Error Detection, restated). *Let $\theta$ be the current parameters and $\eta > 0$ the learning rate. Define the one-step updates*

$$\theta_{SFT}^+ = \theta - \eta(g_{\langle\text{UNDO}\rangle} + g_e + g_{\mathbf{c}}), \tag{22}$$

$$\theta_{mSFT}^+ = \theta - \eta(g_{\langle\text{UNDO}\rangle} + g_{\mathbf{c}}). \tag{23}$$

*Under Assumptions B.1–B.3, the masked update achieves lower expected $\langle$UNDO$\rangle$ cross-entropy:*

$$\mathbb{E}\left[\mathcal{L}_{\langle\text{UNDO}\rangle}(\theta^+_{mSFT})\right] \leq \mathbb{E}\left[\mathcal{L}_{\langle\text{UNDO}\rangle}(\theta^+_{SFT})\right] - \Gamma_{\langle\text{UNDO}\rangle}, \tag{24}$$

*where $\Gamma_{\langle\text{UNDO}\rangle} \geq 0$ is the improvement gap defined in the proof.*

*Proof.* We define $g_m := g_{\langle\text{UNDO}\rangle} + g_{\mathbf{c}}$, $u_m = -\eta g_m$ and $u_s = -\eta(g_m + g_e)$ for brevity. By applying Lemma B.4 to $f = \mathcal{L}_{\langle\text{UNDO}\rangle}$, there exist remainders $r_m, r_s$ with $|r_m| \leq \frac{L}{2}\|u_m\|^2$ and $|r_s| \leq \frac{L}{2}\|u_s\|^2$ such that:

$$\mathcal{L}_{\langle\text{UNDO}\rangle}(\theta^+_{mSFT}) = \mathcal{L}_{\langle\text{UNDO}\rangle}(\theta) + \langle g_{\langle\text{UNDO}\rangle}, u_m \rangle + r_m, \tag{25}$$

$$\mathcal{L}_{\langle\text{UNDO}\rangle}(\theta^+_{SFT}) = \mathcal{L}_{\langle\text{UNDO}\rangle}(\theta) + \langle g_{\langle\text{UNDO}\rangle}, u_s \rangle + r_s. \tag{26}$$

Subtracting Equation 25 from Equation 26 yields:

$$\mathcal{L}_{\langle\text{UNDO}\rangle}(\theta^+_{SFT}) - \mathcal{L}_{\langle\text{UNDO}\rangle}(\theta^+_{mSFT}) = \langle g_{\langle\text{UNDO}\rangle}, u_s - u_m \rangle + (r_s - r_m) \tag{27}$$

$$= -\eta\langle g_{\langle\text{UNDO}\rangle}, g_e \rangle + (r_s - r_m). \tag{28}$$

Since $|r_s - r_m| \leq |r_s| + |r_m|$, we obtain:

$$\mathcal{L}_{\langle\text{UNDO}\rangle}(\theta^+_{SFT}) - \mathcal{L}_{\langle\text{UNDO}\rangle}(\theta^+_{mSFT}) \geq -\eta\langle g_{\langle\text{UNDO}\rangle}, g_e \rangle - \frac{L\eta^2}{2}\left(\|g_m + g_e\|^2 + \|g_m\|^2\right). \tag{29}$$

Taking expectations and applying Assumption B.2(a) gives:

$$\mathbb{E}[-\langle g_{\langle\text{UNDO}\rangle}, g_e \rangle] \geq \alpha_{\langle\text{UNDO}\rangle}\mathbb{E}[\|g_e\|^2]. \tag{30}$$

Applying Assumption B.3, *i.e.*, $\mathbb{E}[\|g_m\|^2 + \|g_m + g_e\|^2] \leq C\mathbb{E}[\|g_e\|^2]$, we obtain:

$$\mathbb{E}\left[\mathcal{L}_{\langle\text{UNDO}\rangle}(\theta^+_{SFT}) - \mathcal{L}_{\langle\text{UNDO}\rangle}(\theta^+_{mSFT})\right] \geq \left(\eta\alpha_{\langle\text{UNDO}\rangle} - \frac{L\eta^2 C}{2}\right)\mathbb{E}\left[\|g_e\|^2\right] = \Gamma_{\langle\text{UNDO}\rangle}. \tag{31}$$

This completes the proof. $\qquad\square$

If the learning rate satisfies $\eta \leq \frac{2\alpha_{\langle\text{UNDO}\rangle}}{LC}$ with $\alpha_{\langle\text{UNDO}\rangle} > 0$, then $\Gamma_{\langle\text{UNDO}\rangle} \geq 0$, yielding the desired result. Since $\mathcal{L}_{\langle\text{UNDO}\rangle}(\theta) = -\log p_\theta(\langle\text{UNDO}\rangle \mid \mathbf{x}, \mathbf{p}, \mathbf{e})$, lower expected cross-entropy corresponds to higher expected log-probability of $\langle$UNDO$\rangle$ under mSFT.

### B.3. Proof of Proposition 2.1 Part (b)

**Proposition B.6** (Corrective Generation, restated). *Let $\mathcal{L}_{\mathbf{c}}(\theta) = -\log p_\theta(\mathbf{c} \mid \mathbf{x}, \mathbf{p}, \mathbf{e}, \mathbf{b})$ denote the correction cross-entropy. Under Assumptions B.1–B.3, the masked update achieves lower expected correction cross-entropy:*

$$\mathbb{E}\left[\mathcal{L}_{\mathbf{c}}(\theta^+_{mSFT})\right] \leq \mathbb{E}\left[\mathcal{L}_{\mathbf{c}}(\theta^+_{SFT})\right] - \Gamma_{\mathbf{c}}, \tag{32}$$

*where $\Gamma_{\mathbf{c}} \geq 0$ is the improvement gap.*

*Proof.* The proof follows the same structure as Part (a), replacing the $\langle$UNDO$\rangle$ loss with the correction loss. Applying Lemma B.4 to $f = \mathcal{L}_{\mathbf{c}}$ with $u_m = -\eta g_m$ and $u_s = -\eta(g_m + g_e)$ gives:

$$\mathcal{L}_{\mathbf{c}}(\theta^+_{SFT}) - \mathcal{L}_{\mathbf{c}}(\theta^+_{mSFT}) = -\eta\langle g_{\mathbf{c}}, g_e \rangle + (r_s - r_m), \tag{33}$$

with $|r_s| \leq \frac{L}{2}\|u_s\|^2$ and $|r_m| \leq \frac{L}{2}\|u_m\|^2$. Using $|r_s - r_m| \leq |r_s| + |r_m|$:

$$\mathcal{L}_{\mathbf{c}}(\theta^+_{SFT}) - \mathcal{L}_{\mathbf{c}}(\theta^+_{mSFT}) \geq -\eta\langle g_{\mathbf{c}}, g_e \rangle - \frac{L\eta^2}{2}\left(\|g_m + g_e\|^2 + \|g_m\|^2\right). \tag{34}$$

Taking expectations and applying Assumption B.2(b) gives $\mathbb{E}[-\langle g_{\mathbf{c}}, g_e \rangle] \geq \alpha_{\mathbf{c}}\mathbb{E}[\|g_e\|^2]$, while Assumption B.3 provides $\mathbb{E}[\|g_m\|^2 + \|g_m + g_e\|^2] \leq C\mathbb{E}[\|g_e\|^2]$. Therefore:

$$\mathbb{E}\left[\mathcal{L}_{\mathbf{c}}(\theta^+_{SFT}) - \mathcal{L}_{\mathbf{c}}(\theta^+_{mSFT})\right] \geq \left(\eta\alpha_{\mathbf{c}} - \frac{L\eta^2 C}{2}\right)\mathbb{E}\left[\|g_e\|^2\right] = \Gamma_{\mathbf{c}}. \tag{35}$$

This completes the proof. $\qquad\square$

Again, if $\eta \leq \frac{2\alpha_{\mathbf{c}}}{LC}$, then $\Gamma_{\mathbf{c}} \geq 0$, yielding the desired result. Lower expected correction cross-entropy corresponds to higher expected log-probability of generating correct continuations under mSFT.

## B.4. Proof of Proposition 2.1 Part (c)

**Proposition B.7** (Reduced Negative Learning, restated). *Let $s_\theta := \log p_\theta(\mathbf{e} \mid \mathbf{x}, \mathbf{p})$ denote the error-score, and define $L_e(\theta) = -s_\theta$ (the error cross-entropy) with gradient $g_e = \nabla_\theta L_e(\theta)$. Under Assumptions B.1–B.3, SFT increases the expected error-score more than mSFT:*

$$\mathbb{E}\left[s(\theta_{SFT}^+)\right] \geq \mathbb{E}\left[s(\theta_{mSFT}^+)\right] + \Gamma_e, \tag{36}$$

*where $\Gamma_e \geq 0$ is the negative learning gap.*

*Proof.* Define $g_m = g_{\langle\text{UNDO}\rangle} + g_{\mathbf{c}}$, with $\theta_{\text{mSFT}}^+ = \theta - \eta g_m$ and $\theta_{\text{SFT}}^+ = \theta - \eta(g_m + g_e)$. Applying Lemma B.4 to $f = s_\theta$ (the error-score), with $u_m = -\eta g_m$ and $u_s = -\eta(g_m + g_e)$, there exist remainders satisfying:

$$s(\theta_{\text{mSFT}}^+) = s(\theta) + \langle\nabla s(\theta), u_m\rangle + r_m, \tag{37}$$

$$s(\theta_{\text{SFT}}^+) = s(\theta) + \langle\nabla s(\theta), u_s\rangle + r_s, \tag{38}$$

with $|r_m| \leq \frac{L}{2}\|u_m\|^2$ and $|r_s| \leq \frac{L}{2}\|u_s\|^2$. Subtracting gives:

$$s(\theta_{\text{SFT}}^+) - s(\theta_{\text{mSFT}}^+) = \langle\nabla s(\theta), -\eta g_e\rangle + (r_s - r_m). \tag{39}$$

Since $L_e(\theta) = -s(\theta)$, we have $\nabla s(\theta) = -g_e$, so $\langle\nabla s(\theta), -\eta g_e\rangle = \eta\|g_e\|^2$. Using $|r_s - r_m| \leq |r_s| + |r_m|$:

$$s(\theta_{\text{SFT}}^+) - s(\theta_{\text{mSFT}}^+) \geq \eta\|g_e\|^2 - \frac{L\eta^2}{2}\left(\|g_m + g_e\|^2 + \|g_m\|^2\right). \tag{40}$$

Taking expectations and applying Assumption B.3 gives $\mathbb{E}[\|g_m\|^2 + \|g_m + g_e\|^2] \leq C\mathbb{E}[\|g_e\|^2]$. Therefore:

$$\mathbb{E}\left[s(\theta_{\text{SFT}}^+) - s(\theta_{\text{mSFT}}^+)\right] \geq \left(\eta - \frac{L\eta^2 C}{2}\right)\mathbb{E}\left[\|g_e\|^2\right] = \Gamma_e. \tag{41}$$

This completes the proof. $\square$

If $\eta \leq \frac{2}{LC}$, then $\Gamma_e \geq 0$, yielding the desired result. This shows that SFT increases the expected log-probability of error tokens more than mSFT does, confirming that masking reduces negative learning.

## C. Token Initialization

We investigate three strategies for initializing the embedding of the special token $\langle\text{UNDO}\rangle$, each encoding different priors about its intended functionality.

**Centroid Initialization.** A baseline approach that initializes $\langle\text{UNDO}\rangle$ as the centroid of the existing vocabulary embedding space:

$$e_{\langle\text{UNDO}\rangle} = \frac{1}{|\mathcal{V}|}\sum_{w\in\mathcal{V}} e_w, \tag{42}$$

where $\mathcal{V}$ denotes the vocabulary and $e_w$ the embedding of token $w$. This initialization makes no assumption about the token's semantics, placing it at a neutral position in the embedding space.

**Context-Based Initialization.** We leverage the model's own representational capacity by encoding a natural language description of the token's functionality (*i.e.*, "this token is used to delete the previous token in the response") and extracting the resulting hidden state as the initialization. This approach grounds the token embedding in the model's pretrained semantic space.

**Semantic Anchor Initialization.** We identify a set of $K$ lexical anchors semantically aligned with the undo operation:

$$\mathcal{W} = \{\texttt{delete}, \texttt{remove}, \texttt{undo}, \texttt{erase}, \texttt{back}, \texttt{cancel}, \texttt{retry}, \texttt{revert}, \texttt{reset}, \texttt{clear}, \texttt{backspace}\}. \tag{43}$$

The token embedding is initialized as the mean of these anchor embeddings:

$$e_{\langle\text{UNDO}\rangle} = \frac{1}{K} \sum_{i=1}^{K} e_{w_i}, \quad w_i \in \mathcal{W}. \tag{44}$$

This strategy encodes an explicit semantic prior, positioning the new token proximal to existing tokens with related meaning.

## D. Additional Reward Functions

In Table 1c, we explored several reward function designs from existing works (Sel et al., 2025; Kumar et al., 2025; Ding et al., 2025). In this section, we discuss the implementation detail of these reward function designs.

- **BSAFE+** (Sel et al., 2025). BSAFE+ primarily addresses safety violations in autoregressive sequence models. Specifically, they introduce a category token `[CATEGORY]` to classify the type of violation, and a backtrack token `[BACKTRACK_BY_X]` that indicates the number of tokens X to retract preceding the backtrack token. Due to these mechanistic differences, we adapt their reward function to align with our approach. Specifically, if no $\langle\text{UNDO}\rangle$ token exists in the generated sequence, we assign $\mathcal{R}(\gamma, \mathbf{y}^*) = +1.0$ when $\text{acc}(\hat{\gamma}, \hat{\mathbf{y}^*}) = 1.0$, and $\mathcal{R}(\gamma, \mathbf{y}^*) = -1.0$ otherwise. When a $\langle\text{UNDO}\rangle$ token is present, we employ a critic model (GPT-4o) to judge whether the backtracking action is warranted. If deemed necessary, we assign $\mathcal{R}(\gamma, \mathbf{y}^*) = +1.0$; otherwise, $\mathcal{R}(\gamma, \mathbf{y}^*) = -0.5$.
- **SCoRe** (Kumar et al., 2025). SCoRe proposes a method to enable language models to self-correct their reasoning through a two-stage reinforcement learning framework. Specifically, they first train the model to improve its second-attempt responses conditioned on self-generated first attempts, then jointly optimize both attempts to prevent reward hacking. Due to the lack of an official source code, we adopt code from the research community[1]. The reward function is defined as follows:

$$\mathcal{R} = \begin{cases} +1.0, & \text{if } \text{acc}(\hat{\tau}, \hat{\mathbf{y}}^*) = 1.0, \\ -1.0 & \text{if } \text{acc}(\hat{\tau}, \hat{\tau}') = 1.0, \\ +0.5, & \text{otherwise}, \end{cases} \tag{45}$$

where, $\tau'$ denotes the response from previous epoch, and $\tau$ is the response in current epoch. Moreover, $\hat{\cdot}$ denotes the post-processed sequence we discussed in Section 2.4.

- **MGRPO** (Ding et al., 2025). MGRPO introduces a two layer hierarchical reinforcement learning framework to enhance the reasoning capabilities of language models. We employ their second layer reward function to optimize the policy model, which is defined as follows:

$$\mathcal{R} = \begin{cases} +1.0, & \text{if } \text{acc}(\hat{\tau}, \hat{\mathbf{y}}^*) > \text{acc}(\hat{\tau}', \hat{\mathbf{y}}^*), \\ +1.0, & \text{if } \text{acc}(\hat{\tau}, \hat{\mathbf{y}}^*) = 1.0 \text{ and } \text{acc}(\hat{\tau}', \hat{\mathbf{y}}^*) = 1.0, \\ -1.0, & \text{otherwise}. \end{cases} \tag{46}$$

Again $\tau'$ denotes the response from previous epoch, and $\tau$ is the response in current epoch, and $\hat{\cdot}$ denotes the post-processed sequence we discussed in Section 2.4.

## E. Additional Sequence Augmentations

In this section, we present two additional sequence augmentation methods discussed in Section 2.1.

### E.1. Stochastic Error Injection

We randomly insert error tokens into the sequence, followed by same amount of backtrack token $\langle\text{UNDO}\rangle$ and the correct sequence. Given a reference sequence $y = (y_1, \ldots, y_n)$, we sample an insertion position $i \in \{1, \ldots, n\}$ and error length $k \in \{1, \ldots, k_{\max}\}$. We sample tokens $e = (e_1, \ldots, e_k)$ as error tokens, where each token $e_j$ is sampled independently from the vocabulary $\mathcal{V}$:

$$e_j \sim \text{Uniform}(\mathcal{V}), \quad j \in \{1, \ldots, k\}. \tag{47}$$

---

[1]https://github.com/daje0601/Google_SCoRe, https://github.com/BY571/SCoRe

The augmented sequence $y^*$ is constructed by inserting the error tokens followed by $k$ backtrack tokens at position $i$:

$$y^* = (y_1, \ldots, y_i, \overbrace{\underbrace{e_1, \ldots, e_k}_{\text{Error Sequence}}, \underbrace{\langle\text{UNDO}\rangle, \ldots, \langle\text{UNDO}\rangle}_{k \text{ backtrack}}}^{\text{Inserted Tokens}}, y_{i+1}, \ldots, y_n). \tag{48}$$

In practice, the procedure generalizes to multiple insertion points. Let $\mathcal{I} = \{(i_1, k_1), \ldots, (i_L, k_L)\}$ denote a set of insertion positions and corresponding error lengths, with $i_1 < i_2 < \cdots < i_L$. The augmented sequence becomes:

$$y^* = (y_1, \ldots, y_{i_1}, \overbrace{\underbrace{e_1^{(1)}, \ldots, e_{k_1}^{(1)}}_{\text{Error Sequence } e^{(1)}}, \underbrace{\langle\text{UNDO}\rangle, \ldots, \langle\text{UNDO}\rangle}_{k_1 \text{ times}}}^{\text{Inserted Tokens at } i_1}, y_{i_1+1}, \ldots, y_{i_2}, \overbrace{\underbrace{e_1^{(2)}, \ldots, e_{k_2}^{(2)}}_{\text{Error Sequence } e^{(2)}}, \underbrace{\langle\text{UNDO}\rangle, \ldots, \langle\text{UNDO}\rangle}_{k_2 \text{ times}}}^{\text{Inserted Tokens at } i_2}, \ldots, y_n) \tag{49}$$

where $e^{(j)} = (e_1^{(j)}, \ldots, e_{k_j}^{(j)})$ are independently sampled error sequences.

### E.2. Hard Sample Sequence Augmentation

Unlike stochastic error injection that samples random tokens from the vocabulary, hard sample sequence augmentation introduces plausible but incorrect tokens that represent realistic reasoning errors. Given a reference sequence $\mathbf{y} = (y_1, \ldots, y_n)$ derived from a symbolic template $\mathcal{T}$, we construct an alternative sequence $\tilde{\mathbf{y}} = (\tilde{y}_1, \ldots, \tilde{y}_m)$ from a different instantiation $\tilde{\mathcal{T}}$ of the same template with modified symbolic values (*e.g.*, different numerical constants or variable names), following GSM-Symbolic (Mirzadeh et al., 2025). We first compute the maximal matching blocks between $\mathbf{y}$ and $\tilde{\mathbf{y}}$:

$$\mathcal{M}(\mathbf{y}, \tilde{\mathbf{y}}) = \{(a_i, b_i, \ell_i)\}_{i=1}^K, \text{ s.t. } y_{a_i:a_i+\ell_i} = \tilde{y}_{b_i:b_i+\ell_i}. \tag{50}$$

For each divergence point $i$ where $\mathbf{y}$ and $\tilde{\mathbf{y}}$ differ, we extract the hard error span $\tilde{\mathbf{e}}_i = (\tilde{y}_{b_i+\ell_i}, \ldots, \tilde{y}_{b_{i+1}-1})$ from $\tilde{\mathbf{y}}$ and the corresponding correct span $\mathbf{c}_i = (y_{a_i+\ell_i}, \ldots, y_{a_{i+1}-1})$ from $\mathbf{y}$. The augmented sequence $\mathbf{y}^*$ is constructed by inserting the hard error tokens followed by tokens $\langle\text{UNDO}\rangle$ at each divergence point:

$$\mathbf{y}^* = (y_1, \ldots, y_{a_i+\ell_i-1}, \overbrace{\underbrace{\tilde{y}_{b_i+\ell_i}, \ldots, \tilde{y}_{b_{i+1}-1}}_{\text{Hard Error Span } \tilde{\mathbf{e}}_i}, \underbrace{\langle\text{UNDO}\rangle, \ldots, \langle\text{UNDO}\rangle}_{|\tilde{\mathbf{e}}_i| \text{ backtracks}}}^{\text{Inserted Tokens}}, \underbrace{y_{a_i+\ell_i}, \ldots, y_{a_{i+1}-1}}_{\text{Correction } \mathbf{c}_i}, \ldots, y_n). \tag{51}$$

In practice, the procedure generalizes to multiple divergence points. Let $\mathcal{D} = \{(a_1, b_1, \ell_1), \ldots, (a_L, b_L, \ell_L)\}$ denote the set of matching blocks between $\mathbf{y}$ and $\tilde{\mathbf{y}}$. For each gap between consecutive blocks, we insert the corresponding hard error span from $\tilde{\mathbf{y}}$:

$$\mathbf{y}^* = (y_1, \ldots, y_{a_1+\ell_1-1}, \overbrace{\tilde{\mathbf{e}}_1, \underbrace{\langle\text{UNDO}\rangle, \ldots,}_{|\tilde{\mathbf{e}}_1| \text{ times}}}^{\text{Divergence 1}} \mathbf{c}_1, \ldots, y_{a_2+\ell_2-1}, \overbrace{\tilde{\mathbf{e}}_2, \underbrace{\langle\text{UNDO}\rangle, \ldots,}_{|\tilde{\mathbf{e}}_2| \text{ times}}}^{\text{Divergence 2}} \mathbf{c}_2, \ldots, y_n), \tag{52}$$

where $\tilde{\mathbf{e}}_j$ and $\mathbf{c}_j$ denote the hard error span and correction span at divergence point $j$, respectively. The key distinction from stochastic error injection is that hard error tokens are semantically coherent within the problem context, representing the type of plausible mistakes a model might generate during reasoning, *e.g.*, using a wrong intermediate value or applying an incorrect operation.

## F. Result Examples

*Table 7.* Example of N-MARS generation on GSM8K, that does not require erase actions.

| Question | A robe takes 2 bolts of blue fiber and half that much white fiber. How many bolts in total does it take? Let's think step by step. Put your final answer at the end with 'The answer is: .' |
|---|---|
| Response | The white fiber takes half the amount of the blue fiber which means $1/2 \times 2 = 1$ bolt. So the total number of bolts is $2 + 1 = 3$ bolts. The answer is 3. |
| Target Answer | 3 |

*Table 8.* Example of N-MARS generation on GSM8K that requires erase actions.

| Question | John drives for 3 hours at a speed of 60 mph and then turns around because he realizes he forgot something very important at home. He tries to get home in 4 hours but spends the first 2 hours in standstill traffic. He spends the next half-hour driving at a speed of 30mph, before being able to drive the remaining time of the 4 hours going at 80 mph. How far is he from home at the end of those 4 hours? Let's think step by step. Put your final answer at the end with 'The answer is: .' |
|---|---|
| Response | He spends 4 hours driving at 30mph⟨UNDO⟩ ⟨UNDO⟩ ⟨UNDO⟩, but 2 hours are in standstill traffic. So that leaves 4-2 = 2 hours. He spends half of 2 hours at 30mph on trips so he has 30*.5 = 15 miles. He has 80 mph for the remaining 4 hours⟨UNDO⟩ ⟨UNDO⟩2-.5 =1.5 hours, so that is 80*1.5 = 120 miles. All total, he has driven 120 + 15 = 135 miles. Since he is 60*3 = 180 miles away from home initially, he is 180 - 135 = 45 miles away from home. The answer is 45. |
| Target Answer | 45 |

