# OpenReview forum: "Non-Monotonic Autoregressive Sequence Model"
_ICML.cc/2026/Conference — ICML 2026 regular_

### Official Review · Reviewer_fZNn · 2026-02-25

**Soundness:** 3
**Presentation:** 3
**Significance:** 3
**Originality:** 3
**Overall Recommendation:** 5
**Confidence:** 4

**Summary:**

This paper proposes N-MARS, a non-monotonic autoregressive sequence modeling framework that enables a model to revise earlier tokens during a single decoding pass via a learned erase action token ⟨UNDO⟩. The training pipeline consists of (i) sequence alignment augmentation that converts model deviations into error-correction traces by inserting error spans followed by the same number of ⟨UNDO⟩ tokens and then the ground-truth continuation, (ii) masked supervised fine-tuning (mSFT) that masks loss on the injected error tokens to avoid “negative learning” while still conditioning on them as context, and (iii) GRPO-based reinforcement learning with a reward that combines final correctness and a penalty/bonus term that regulates ⟨UNDO⟩ usage. Experiments on GSM8K, MATH, and MBPP across several backbones (e.g., Llama-3.x and Qwen) show consistent gains over SFT/RFT/STaR+ and a multi-pass revision baseline (ReVISE), with ablations analyzing token initialization, augmentation strategies, reward components, and κ sensitivity.

**Compliance With Llm Reviewing Policy:**

Affirmed.

**Final Justification:**

My final recommendation is Accept. I find the paper original, technically solid, and clearly presented. The core idea, introducing an explicit ⟨UNDO⟩ action so an autoregressive model can revise tokens during decoding, is novel and well motivated. The method is supported by a coherent training pipeline (sequence augmentation, masked SFT, and GRPO) and a fairly thorough empirical evaluation across multiple models and reasoning benchmarks, with useful ablations. The main remaining weakness is that the evaluation is still focused on verifiable reasoning tasks, and long-context behavior is discussed more as a limitation than directly tested.

The rebuttal addressed my main concerns and improved my assessment. In particular, the added compute-matched comparisons and latency/token-overhead statistics make the efficiency claims more convincing and reduce my concern that the gains are mainly due to extra compute. The authors also gave a reasonable clarification of the long-context limitation and possible extensions. Overall, the rebuttal changed my view from a cautious Weak Accept to Accept.

**Key Questions For Authors:**

1. Did you compare N-MARS to standard SFT (or ReVISE-style baselines) under the same inference compute budget (e.g., same maximum generated tokens / same FLOPs / same wall-clock), including allowing the baseline to emit longer reasoning traces or “pause/deliberation” tokens? If not, can you add a controlled study to rule out “extra compute” as the main driver of gains?

2. What is the measured latency/throughput impact of ⟨UNDO⟩ at inference (e.g., tokens generated per solved instance, wall-clock per sample) across datasets and κ settings? How often does the model backtrack, and what is the distribution of ⟨UNDO⟩ counts on test data?

3. Since the raw trajectory (including error tokens and ⟨UNDO⟩) remains part of the context, what are the expected challenges when applying N-MARS to long-context generation (multi-turn dialogue, long-form writing)? Do you anticipate needing higher-level edit actions (sentence/segment-level undo) or memory/position handling changes to avoid context bloat?

**Limitations:**

yes

**Strengths And Weaknesses:**

Pros:

1. Implementing backtracking via an explicit ⟨UNDO⟩ token and stack-style post-processing is simple, model-agnostic, and easy to integrate into standard decoding pipelines.

2. The paper clearly identifies the “train on error tokens → imitate errors” pitfall and proposes mSFT to address it, supported by a theoretical argument and empirical evidence that SFT on augmented traces can degrade performance while mSFT improves it.

3. Results span multiple backbones and multiple reasoning domains (math + code), with thoughtful design-space exploration (token initialization, reward variants, augmentation variants, κ sweeps, test-time scaling).


Cons:

1. Because N-MARS can generate extra tokens (error spans + ⟨UNDO⟩ + regenerated content), improvements may partially reflect more test-time compute, but the paper does not present compute-matched comparisons (e.g., fixed total token budget across methods) or wall-clock/throughput reporting.

2. The approach operates at token-level revision and the raw trajectory remains in-context, which may exacerbate cost and context-window pressure for long-horizon tasks (dialogue, summarization, tool traces). This is acknowledged only qualitatively, without dedicated experiments.

3. The RL reward uses a correctness signal (rule-based verifier in the current setup), which is natural for GSM8K/MATH/MBPP but may not transfer cleanly to open-ended generation without strong verifiers or preference models; the paper could clarify how robust the method is under weaker feedback.

---

> ### Author Rebuttal · Authors · 2026-03-31
>
> > **Q1 / W1: Compute-matched comparisons.**
>
> We have conducted additional compute-matched experiments (Num of new tokens = {128, 256, 512}) as shown below.
>
> | Dataset   | Method | 128 | 256 | 512 |
> |-----------|--------|-----|-----|-----|
> | GSM8K     | Base   | 5.1   | 8.8 | 8.8 |
> | GSM8K     | ReVISE | 27.2| 28.1 | 28.1 |
> | GSM8K     | N-MARS | **29.6** | **31.3** | **31.3** |
> | MATH-500  | Base    | 2.4 |  3.2 |  3.2|
> | MATH-500  | ReVISE | 12.0 | 13.4 | 13.4 |
> | MATH-500  | N-MARS | **13.1** | **15.2** | **15.9** |
>
>
> According to the table, we find that, at the same token budget, N-MARS consistently outperforms ReVISE and SFT on both datasets, indicating that the performance gain of N-MARS is not solely due to more generated tokens but also to the more effective use of the UNDO mechanism.
>
> > **Q2: Measured latency and $\langle \text{UNDO}\rangle$ statistics on test set.**
>
> We have measured the latency, average new tokens generated, and $\langle \text{UNDO}\rangle$ statistics on the test set, as shown in the tables below.
>
> | **Dataset** | **Method** | **Acc** $\uparrow$ | **Avg. New Tokens** $\downarrow$ | **Latency (mins)** $\downarrow$|
> | :--- | :--- | :--- | :--- | :--- |
> | GSM8K | Base | 8.8 | **118.3** | **7.2** |
> | GSM8K | ReVISE | 28.1 | 178.9 | 9.8 |
> | GSM8K | **N-MARS** | **31.3** | 153.7 | 9.1 |
> | MATH-500 | Base | 3.2 | **174.4** | **9.4** |
> | MATH-500 | ReVISE | 13.4 | 249.3 | 11.7 |
> | MATH-500 | N-MARS | **15.2** | 217.5 | 10.4 |
>
>   | Metric | GSM8K | MATH-500 |
>   |--------|-------|----------|
>   | Samples with ≥1 UNDO | 40.6% | 52.2% |
>   | Avg UNDO / sample (all) | 4.8 | 8.4 |
>   | Avg UNDO / sample (with UNDO) | 11.8 | 16.1 |
>
> According to the tables, we find that N-MARS achieves a better accuracy than ReVISE while generating fewer new tokens and having a lower latency.
> Moreover, we observe that a significant portion of samples (40.6% for GSM8K and 52.2% for MATH-500) involve at least one $\langle \text{UNDO}\rangle$ action, and the majority of corrections are achieved with a small number (~11 - 16) of $\langle \text{UNDO}\rangle$ tokens, demonstrating that the model learns to detect and correct errors efficiently via the $\langle \text{UNDO}\rangle$ mechanism.
>
> > **Q3 / W2 : Expected challenges for long-context generation.**
>
> We acknowledge in the Limitation section that the computational overhead scales with long-context generation.
> Two natural extensions we plan to explore:
>
> * Coarser-granularity edit actions, such as sentence-level or chunk-level $\langle \text{UNDO}\rangle$, which can retract larger spans of text with fewer tokens.
> For example, instead of multiple $\langle \text{UNDO}\rangle$, a single sentence-level $\langle \text{UNDO}\rangle$  to retract entire sentences can be more efficient for long-context tasks.
> * Compressing/collapsing the error portion of the context after correction to recover the context window, which can alleviate the context bloat issue.
>
> However, we emphasize that the current study positions itself as a simple and model-agnostic framework that enables token-level revision via $\langle \text{UNDO}\rangle$.
> We will discuss more details about the aforementioned long-context challenges in our revision, and we view these potential extensions as our future work.
>
> > **W3: Transferability to open-ended generation without strong verifiers.**
>
> Our reward design can be extended to open-ended generation by replacing the rule-based verifier with weaker reward signals, such as a learned reward model or LLM-as-judge reward.
> Table 1(c) in our original manuscript already reports performance when replacing our rule-based verifier with the reward design from BSAFE+, which employs a **critic model** (GPT-4o) as a judge to provide reward signals.
> We find that the rule-based verifier (N-MARS) and weak reward signals (from BSAFE+) achieve the same performance over MATH-500 and comparable performance over GSM8K. This demonstrates that N-MARS can leverage weaker reward signals for complex reasoning generation tasks and has potential for broader tasks like open-ended generation, with the trade-off being increased computational cost for the reward model.

---

> > ### Author Rebuttal · Reviewer_fZNn · 2026-04-06
> >
> > My main concerns in the original review were whether the reported gains could largely be explained by extra test-time compute, what the latency/token overhead of the ⟨UNDO⟩ mechanism looks like in practice, and how the approach might behave in longer-context settings. The rebuttal substantially addresses the first two points by adding compute-matched comparisons at fixed token budgets and by reporting latency, average new-token counts, and ⟨UNDO⟩ usage statistics. These additions materially strengthen the paper and address the central fairness/efficiency concern from my review.
> >
> > I also appreciate the authors’ clarification on longer-context limitations. The rebuttal now explicitly acknowledges the context-growth issue and outlines reasonable extensions such as coarser-grained edit actions and compressing corrected spans after revision. In addition, the rebuttal addresses my concern about dependence on strong verifiers by arguing that the framework can also work with weaker reward signals, consistent with the BSAFE+-style comparison already discussed in the manuscript. While long-context and open-ended settings are still not directly evaluated, I view those as limitations and future-work questions rather than blockers for the current paper. Overall, my main concerns have been adequately addressed.

---

### Official Review · Reviewer_DLhi · 2026-03-09

**Soundness:** 2
**Presentation:** 3
**Significance:** 2
**Originality:** 2
**Overall Recommendation:** 4
**Confidence:** 3

**Summary:**

This paper proposes N-MARS, which introduces an `<UNDO>` token so that a language model can retract previously generated tokens
 within a single decoding trajectory. The method combines alignment-based augmentation, mSFT, and GRPO-based RL, and report improvements over several baselines on GSM8K, MATH, and MBPP.

   The paper is well motivated and clearly written, and mSFT is a useful component. However, I have concerns about the novelty of
 the explicit `<UNDO>` mechanism, the fact that the underlying prediction process remains standard left-to-right autoregression, and
 the fairness of the main comparisons given that N-MARS is trained on richer augmented error-correction traces than simpler baselines such as standard SFT.

**Compliance With Llm Reviewing Policy:**

Affirmed.

**Key Questions For Authors:**

1. Why is an explicit <UNDO> token preferable to natural-language self-correction? Please compare against a strong baseline that uses the same augmented training traces but replaces <UNDO> with natural-language revision markers.

 2. In what precise sense is the method non-monotonic? Since the model still performs left-to-right autoregressive prediction over the edit trajectory, would it be more accurate to describe the method as autoregressive generation over editable trajectories rather than truly non-autoregressive or fundamentally non-monotonic generation?

 3. How fair are the comparisons in Table 2? Since N-MARS is trained on richer augmented traces, can the authors provide data-matched baselines where SFT/ReVISE-style methods are trained with the same amount and type of synthetic supervision?

 4. What is the actual efficiency cost of the method? Please report average generated tokens, average number of <UNDO> tokens, average correction depth, and wall-clock latency, ideally compared with multi-pass baselines under matched compute budgets.

 5. Does the method generalize beyond tasks with strong verification signals? The current experiments focus mainly on math and code benchmarks where correctness can be measured relatively cleanly. How would the method extend to more open-ended generation tasks?

**Limitations:**

The paper has several limitations that currently restrict its impact.

 First, the core mechanism depends on a newly introduced control token whose semantics are not part of standard pretraining. This
 raises the question of whether the method is leveraging the model’s natural language ability in the most effective way, or instead
 imposing an artificial editing protocol at post-training time.

 Second, the method is only “non-monotonic” at the level of the post-processed visible output. The underlying prediction process
 remains fully autoregressive and linear. As a result, the approach may not address the deeper limitation of autoregressive reasoning as strongly as the paper suggests.

 Third, the current empirical evidence does not cleanly disentangle method improvements from data/supervision improvements, since the model is trained on richer augmented traces than the simpler baselines. This makes it difficult to determine how much of the gain comes from the <UNDO> mechanism itself.

 Fourth, the method is evaluated mostly on verifiable reasoning tasks such as GSM8K, MATH, and MBPP. It remains unclear whether the same approach would be effective in settings where correctness is ambiguous and reward design is less straightforward.

**Strengths And Weaknesses:**

## Strengths

1. **Important problem.**
   The paper trys to address error propagation in autoregressive language models, which is a meaningful limitation for reasoning tasks.

2. **Clean training recipe.**
   The overall pipeline is coherent, and **mSFT** is a useful component for avoiding negative learning on augmented traces.

3. **Generally consistent gains.**
   The method improves over the reported baselines across multiple models and tasks.

## Weaknesses

1. **The paper does not establish that explicit `<UNDO>` tokens are preferable to natural-language self-correction.**
   A major missing baseline is a data-matched verbal-reflection variant, where the model is trained on the same augmented traces but uses natural-language revision patterns (e.g., “that was incorrect, let me revise”) instead of a newly introduced control token. Without such a comparison, it is hard to know whether the explicit `<UNDO>` mechanism is truly necessary or simply one possible implementation.

2. **The claimed non-monotonicity is limited to the visible output sequence, not the underlying prediction process.**
   The model still performs standard left-to-right next-token prediction over an edit-action trajectory. In that sense, the method is still autoregressive at the level of actual computation; only the reduced final sequence appears non-monotonic after applying the stack-like post-processing rule. This makes the conceptual claim somewhat weaker than it initially appears.

3. **The main comparisons may not be fully fair.**
   N-MARS is trained on augmented error-correction traces, which provide richer supervision than the training data used by standard baselines such as SFT, and arguably a different kind of synthetic signal than RFT/STaR+-style methods. As a result, Table 2 does not cleanly isolate the benefit of the `<UNDO>` mechanism itself; it instead compares full recipes with different supervision richness.

4. **Efficiency claims are under-supported.**
   One motivation of the paper is that integrating correction into a single trajectory could be more efficient than post-hoc verification or multi-pass refinement. However, the paper does not provide direct evidence such as latency, total generated tokens, `<UNDO>` frequency, or matched-compute comparisons against methods like ReVISE.

---

> ### Author Rebuttal · Authors · 2026-03-31
>
> We thank the reviewer for the careful and detailed reading. We address each concern directly.
>
> > **Q1 / W1: Why $\langle \text{UNDO} \rangle$ is preferable to natural-language self-correction?**
>
> Compared to NL-based self-reflection, N-MARS generates a **clean and corrected** output by design.
> However, an NL-based method needs to learn **the error starting position**, in order to remove the error tokens; otherwise, the error tokens will remain in the generated sequence.
>
> To directly address the reviewer's concern, we evaluate a verbal-reflection baseline (denoted as Self-Reflect) using the same augmented training traces:
>
> | Backbone | Method  | GSM8K | MATH-500 |
> | --- | --- | --- | --- |
> | Llama-3.2-1B | Self-Reflect |  25.8 | 11.6 |
> | Llama-3.2-1B | N-MARS | **31.3** |  **15.2**|
> | Llama-3.1-8B | Self-Reflect |  60.9 | 34.5|
> | Llama-3.1-8B | N-MARS | **63.4** |  **36.2**|
>
> N-MARS consistently outperforms Self-Reflect, indicating the necessity of an explicit $\langle \text{UNDO} \rangle$ mechanism.
>
> > **Q2 / W2: In what precise sense is N-MARS non-monotonic?**
>
> Thank you for pointing this out. We **do not intend to claim** our method is **non-autoregressive**; indeed, as the name Non-Monotonic Autoregressive Sequence Model (N-MARS) suggests, it remains built upon the autoregressive decoding paradigm. However, the visible output, i.e., produced via the deterministic stack-based operator, is constructed non-monotonically, as illustrated in Figure 1.
> Our goal is to introduce a **simple and easy-to-integrate method** into the **standard autoregressive decoding paradigm** that enables models to revise their committed outputs, resulting in a **non-monotonic generation** at the surface level.
>
> > **Q3 / W3: N-MARS trained on richer augmented traces.**
>
> We would like to clarify that the augmented trace generation (Section 2.1) is part of our proposed method, not an external data advantage.
> This is analogous to RFT, STaR+, and ReVISE, as they also **leverage self-generated augmented data for training**.
>
> We point the reviewer to **Table 1b (SFT)** and **Table 1d (GRPO)** in our original manuscript, where we compare SFT and mSFT with the same augmented data.
> The result shows that the augmented data alone is not beneficial; instead, the ⟨UNDO⟩ mechanism (enabled by mSFT and GRPO together) is what drives improvement.
>
> > **Q4 / W4: Efficiency Analysis of N-MARS.**
>
> We have conducted experiments to analyze the efficiency of N-MARS:
>
> | **Dataset** | **Method** | **Acc** $\uparrow$ | **Avg. New Tokens** $\downarrow$ | **Latency (mins)** $\downarrow$|
> | :--- | :--- | :--- | :--- | :--- |
> | GSM8K | Base | 8.8 | **118.3** | **7.2** |
> | GSM8K | ReVISE | 28.1 | 178.9 | 9.8 |
> | GSM8K | N-MARS | **31.3** | 153.7 | 9.1 |
> | MATH-500 | Base | 3.2 | **174.4** | **9.4** |
> | MATH-500 | ReVISE | 13.4 | 249.3 | 11.7 |
> | MATH-500 | N-MARS | **15.2** | 217.5 | 10.4 |
>
> The results show that N-MARS achieves better performance than ReVISE with fewer generated tokens and lower latency, indicating the efficiency of our proposed method.
>
> | Dataset   | Method | 128 | 256 | 512 |
> |-----------|--------|-----|-----|-----|
> | GSM8K     | Base   | 5.1   | 8.8 | 8.8 |
> | GSM8K     | ReVISE | 27.2| 28.1 | 28.1 |
> | GSM8K     | N-MARS | **29.6** | **31.3** | **31.3** |
> | MATH-500  | Base    | 2.4 |  3.2 |  3.2|
> | MATH-500  | ReVISE | 12.0 | 13.4 | 13.4 |
> | MATH-500  | N-MARS | **13.1** | **15.2** | **15.9** |
>
> We find that at the same new token budget, i.e., 128, 256, 512, N-MARS consistently outperforms baseline methods, indicating that the performance gain is not due to more generated tokens, but from the effectiveness of the $\langle\text{UNDO}\rangle$ mechanism.
>
>   | Metric | GSM8K | MATH-500 |
>   |--------|-------|----------|
>   | Samples with ≥1 UNDO | 40.6% | 52.2% |
>   | Avg UNDO / sample (all) | 4.8 | 8.4 |
>   | Avg UNDO / sample (with UNDO) | 11.8 | 16.1 |
>
> We observe that a significant portion of samples involve at least one $\langle \text{UNDO}\rangle$ action, and the majority of corrections are achieved with a small number (~11 - 16) of $\langle \text{UNDO}\rangle$ tokens, demonstrating that the model learns to detect and correct errors efficiently via the $\langle \text{UNDO}\rangle$ mechanism.
>
> > **Q5: Does the method generalize beyond verifiable reasoning tasks?**
>
> Our method can be extended to open-ended generation by replacing the rule-based verifier with rewards from metrics, learned preference models, or LLM-as-judge.
> Table 1(c) in our original manuscript already reports performance when replacing our rule-based verifier with the reward design from BSAFE+, i.e., employs a **critic model** (GPT-4o) as a judge to provide reward signals.
> According to the table, N-MARS and BSAFE+ achieve similar performance, indicating the potential of N-MARS for broader generation tasks, such as open-ended generation under weaker reward signals, with the trade-off being increased computational cost for the reward model.

---

> > ### Author Rebuttal · Reviewer_DLhi · 2026-04-01
> >
> > The authors have included additional reasonable experiments and presented supporting results. However, I still have some concerns about the novelty, and therefore I lowered my confidence while raising my score.

---

> > > ### Author Response · Authors · 2026-04-06
> > >
> > > We are glad to see that our response has alleviated the reviewer's concerns.
> > > We would like to further clarify our method's novelty as follows:
> > >
> > > 1. **Masked SFT:** We employ a novel training strategy called mSFT that masks the error tokens during fine-tuning, which is distinct from existing methods that commonly rely on SFT.
> > > Moreover, our proposed mSFT is theoretically supported by Proposition 2.1, which provides a rigorous justification to showing mSFT yields improvements in error detection, corrective generation, and alleviates negative learning, comparing to SFT.
> > >
> > > 2. **Reward Function Design:** We design a novel reward function for GRPO that incorporates an incentive reward for a correct answer and a penalty to reward the usage of the `<UNDO>` token for correction.
> > > Moreover, the penalty reward is controlled by $\kappa$ to encourage early error detection.
> > >
> > > 3. **Backtrack granularity:** Unlike existing method that operates at the reasoning step level (e.g., Self-Backtracking) or full response level (e.g., ReVISE), our method operates at the token level, which is more fine-grained and task-agnostic.
> > > Token-level backtracking aligns with the atomic unit of autoregressive generation, and produces clean outputs without meta-commentary, preserving naturalness.
> > >
> > > We hope our response can alleviate the reviewer's concerns regarding the novelty of our method. Please let us know if you have any further questions or concerns.
> > >
> > >
> > > [1] Step Back to Leap Forward: Self-Backtracking for Symbolic Reasoning and Planning in Language Models. AAAI'26.
> > >
> > > [2] ReVISE: Learning to Refine at Test-Time via Intrinsic Self-Verification. ICML'25

---

### Official Review · Reviewer_cVJs · 2026-03-09

**Soundness:** 3
**Presentation:** 4
**Significance:** 3
**Originality:** 3
**Overall Recommendation:** 5
**Confidence:** 3

**Summary:**

AR models generate sequences in a monotonic manner. That means, once a token is sampled, even if it is incorrect or suboptimal, the model cannot revisit or revise it.

To address this limitation, the authors propose N-MARS, a non-monotonic framework that allows models to generate, evaluate, and revise tokens within a single forward pass, thereby enabling exploration before commitment. The authors insert a learnable < UNDO > token into the vocabulary that retracts the previously generated token, making on-the-fly revision possible within the standard enabling on-the-fly revision within standard autoregressive decoding.

For training, the authors introduce a sequence augmentation strategy, mSFT, and GRPO-based method. Experiments show the benefits of N-MARS with different LLMs on different benchmarks.

**Compliance With Llm Reviewing Policy:**

Affirmed.

**Final Justification:**

Thank the authors for the very detailed and thoughtful responses. My main concerns have been largely addressed, and I appreciate their effort in clarifying the questions.

**Key Questions For Authors:**

Please see the Weaknesses (Major) section.

**Limitations:**

Yes. The authors discussed the limitations and potential societal impact of their work in detail (Page 9).

**Strengths And Weaknesses:**

## Strengths

- N-MARS enables AR models to revisit and refine the tokens previously generated, which is very interesting.

- The proposed method and training strategies are well-motivated and easy to understand.

- Experiments show significant advantages of N-MARs and provide more insights.

## Weaknesses

### Major

- The < UNDO > token erases only 1 token each time. This might not be efficient in some cases. For example, if the model finds some error in the last paragraph in its response, it needs to generate a number of < UNDO > tokens. Even though it is feasible in computation, it seems very hard to train models to do that.

- Lack of efficiency analysis. Most of experiments focus on accuracy on benchmarks, and I didn’t find efficiency reports in the paper. I notice that the authors mentioned in the limitation section “the computational overhead: external verification methods > non-monotonic AR > basic AR”. But I think it would be better to include an efficiency comparison.

- I’m not sure if it is easy to construct “errors” for models to refine. In the paper, the authors propose a way, “identify error indices as positions in \hat{y} not covered by any matching block”. This doesn’t sound very convincing to me, because tokens that are not matched may not be wrong.

Considering that AR has been used in various domains, such as code and images, I feel it’s probably hard to design “errors” in different tasks.

### Minor

- It would be better if the authors could provide more examples or observations on LLM behaviors with N-MARS. I think the < UNDO > token is very interesting, so I’m wondering how LLMs will utilize this token, like how many tokens are used in average, how much portions of < UNDO > really delete errors, and what kind of errors are erased.

---

> ### Author Rebuttal · Authors · 2026-03-30
>
> We appreciate the reviewer's positive assessment and thoughtful questions. We address each concern below.
>
> > **W1: The behavior of $\langle\text{UNDO}\rangle$ might not be efficient in some cases.**
>
> We understand the reviewer's concern that erasing one token at a time may seem inefficient for long error spans.
> However, this token-level granularity is central to our motivation: **Fine-grained correction** within autoregressive models through **non-monotonic generation**.
> Coarser mechanisms, e.g., step-level, sacrifice precision and task-agnostic applicability.
> To further alleviate this, by the reward function design, we introduce a budget κ that encourages **early error detection** during GRPO training, minimizing the need for long $\langle\text{UNDO}\rangle$  sequences at latter generation.
> Moreover, recent interpretability studies [1, 2] demonstrate that LLMs can 'foresee' tokens beyond immediate next-token prediction, enabling early error detection before a long error sequence accumulates.
>
> To empirically validate this, we report the statistic of $\langle\text{UNDO}\rangle$ usage in our experiments.
>   | Metric | GSM8K | MATH-500 |
>   |--------|-------|----------|
>   | Samples with ≥1 UNDO | 40.6% | 52.2% |
>   | Avg UNDO / sample (all) | 4.8 | 8.4 |
>   | Avg UNDO / sample (with UNDO) | 11.8 | 16.1 |
>
> We find that the majority of corrections are achieved with a small number (~11 - 16) of $\langle\text{UNDO}\rangle$ tokens, demonstrating that the model learns to detect and correct errors efficiently.
>
> > **W2: Lack of efficiency analysis.**
>
> We report the efficiency comparison as below:
>
> | **Dataset** | **Method** | **Acc** $\uparrow$  | **Avg. New Tokens** $\downarrow$  | **Latency (mins)** $\downarrow$ |
> | :--- | :--- | :--- | :--- | :--- |
> | GSM8K | Base | 8.8 | **118.3** | **7.2** |
> | GSM8K | ReVISE | 28.1 | 178.9 | 9.8 |
> | GSM8K | N-MARS | **31.3** | 153.7 | 9.1 |
> | MATH-500 | Base | 3.2 | **174.4** | **9.4** |
> | MATH-500 | ReVISE | 13.4 | 249.3 | 11.7 |
> | MATH-500 | N-MARS | **15.2** | 217.5 | 10.4 |
>
> We find that N-MARS achieves better performance and is more efficient than ReVISE, by generating fewer new tokens and incurring lower latency.
>
> > **W3: Error identification via unmatched tokens may not correspond to actual errors.**
>
> We agree that the unmatched tokens may not be incorrect.
> For example, "8 apples" and "eight apples" are semantically equivalent but token-wise different.
> However, this is intentional and actually a feature of our design:
>  1. **mSFT handles noisy labels gracefully:** Even if some "error" tokens are actually valid alternatives, masking them from the loss simply means the model doesn't learn from those positions. This is a conservative choice, i.e., we prefer mask a correct-but-different token than force the model to learn from a genuinely
>   wrong one.
>   3. **The alternative is worse:** Requiring perfect error identification would need an external verifier or human annotation, which may hard to scale up for large datasets and diverse domains, and would introduce its own noise and bias.
>
> Extend N-MARS to broader domains, e.g., CV, would require different error construction methods, as the token-level sequence alignment may not be meaningful for images.
> Some possible directions include utilize spatial coherence metrics or classifier-based error detection. We acknowledge this as nontrivial future work.
>
> > **W4: More examples and observations on `<UNDO>` behavior would be helpful.**
>
> We thank the reviewer for this question. Due to the space limit, we cannot include more examples in the rebuttal.
> We refer the reviewer to Tables 5 and 6 in Appendix, which show two generation examples: (1) a case where N-MARS solves correctly without any `<UNDO>` , and (2) a case where the model emits `<UNDO>` tokens to correct mid-generation arithmetic errors.
> We will provide more examples on $\langle\text{UNDO}\rangle$ behavior in our next revision.
> Empirically, we find that:
> 1. **Local correction:** The model indeed generates $\langle\text{UNDO}\rangle$ tokens to correct errors, and the model detects errors within a few tokens of generation.
> 2. **Arithmetic Errors:** The model mainly emits $\langle\text{UNDO}\rangle$  tokens to correct mid-generation arithmetic errors.
> 3. **Clean output:** After generation, the final sequence reads as a coherent, correct chain of reasoning with no trace of errors.
>
> [1] On the biology of a large language model. Transformer Circuits. Anthropic. (2025).
>
> [2] Tracing the thoughts of a large language model. Anthropic. (2025).

---

> > ### Author Rebuttal · Reviewer_cVJs · 2026-04-01
> >
> > ### Rebuttal Acknowledgment
> >
> > Thank the authors for the detailed and thoughtful responses. My main conerns have been largely addressed after reading both the rebuttal. I have also read the comments from other reviewers.
> >
> > My primary concern was about efficiency, which was also consistently raised by all other reviewers. The additional efficiency comparison provided by the authors is helpful. While the proposed method is slightly slower than the original baseline, it is still faster than ReVISE, and the overall runtime appears acceptable given the performance gains.
> >
> > I also took note of the discussions on novelty raised by Reviewers e2Aq and DLhi. It is true that in the LLM literature, many works explore ideas such as self-improvement, reflection, and iterative reasoning. That said, I find the authors’ clarification regarding the distinction from Self-Backtracking convincing.
> >
> > ### Additional Comments
> >
> > The following comments are forward-looking and do not affect my current evaluation or score. I would like to raise a couple of concerns regarding scalability:
> >
> > W1: Efficiency on more challenging tasks.
> > Datasets such as GSM8K and MATH-500 may not fully reflect the difficulty of more complex reasoning scenarios. As the authors show in the rebuttal, the need for < UNDO > is relatively limited on these benchmarks. However, as pointed out by Reviewer fZNn, it remains unclear whether the efficiency will hold on more challenging datasets. While the authors note that sentence-level < UNDO > could be a possible extension, it may reduce the distinction from Self-Backtracking and other related works.
> >
> > W3: Constructing useful “errors” at scale.
> > The method relies on generating meaningful errors for refinement. It is unclear how easily such errors can be constructed in large-scale or multimodal settings, where defining and detecting “mistakes” may be significantly more challenging.
> >
> > ### Summary
> >
> > In general, the design of the < UNDO > token and the corresponding training strategy are interesting and represent a meaningful contribution. Based on the above, I am inclined to maintain my score.

---

> > > ### Author Response · Authors · 2026-04-06
> > >
> > > We appreciate the reviewer's positive feedback.
> > > For your comments, we would like to address them as follows:
> > >
> > > > **W1: Efficiency on more challenging tasks.**
> > >
> > > We thank the reviewer for this insightful comment.
> > >
> > > First, we believe the current `<UNDO>` mechanism is already efficient compared to the best baseline ReVISE, as shown in the previous rebuttal.
> > >
> > > For more complex reasoning tasks or open-ended generation, it is unnecessary to employ a one-sentence-level `<UNDO>` token to rewind an entire sentence, as it may be hard to define a "sentence" in some cases.
> > > Instead, several design spaces are available to make the mechanism more flexible and efficient.
> > > For example, we can intuitively design a number-aware `<UNDO>` mechanism, i.e., leverage `<UNDO>k ` to backtrack `k` tokens,
> > > where `<UNDO>` is a special token and `k` is a standard number token (e.g., 1, 2, ..., $K_\text{max}$), indicating how many tokens to backtrack.
> > > Mention that the space after the number `k` is necessary to avoid cases where a number token follows `<UNDO>k ` that belongs to the correction part.
> > > This is implemented as a bounded set of macro, i.e., actions in the same MDP, the state remains the current prefix, and the transition deterministically rewinds by $k$ positions.
> > > Training labels for the correct $k$ are directly derived from the sequence alignment procedure, i.e., the length of each error span $|\mathbf{e}_i|$ in Eq. (5) gives the number `k`.
> > > The masking strategy in mSFT extends naturally, with error spans still masked and the `<UNDO>k ` token pair replacing the repeated `<UNDO>` sequence.
> > > In this way, we can prevent generating a large number of `<UNDO>` tokens, especially for long sentences, thereby reducing latency.
> > >
> > > We note that this is a future extension.
> > > The current paper demonstrates that even single-token `<UNDO>` yields significant gains, and the `<UNDO>k ` generalization is a potential direction for scaling to tasks with longer correction spans.
> > >
> > > > **W3: Constructing useful “errors” at scale.**
> > >
> > > We agree that constructing useful errors at scale is more challenging in broader domains.
> > >
> > > Let's take vision autoregressive (AR) models, e.g., LlamaGen [1], as an example for text-to-image generation.
> > > Images are encoded as discrete VQ tokens, and the same alignment procedure applies.
> > > For tolerance to perceptually equivalent tokens, the exact match criterion can be relaxed to a soft match based on codebook embedding distance: $\text{match}(v_i, \hat{v_j}) = \mathbb{1}[\|e_{v_i} - e_{\hat{v_j}}\|_2 < \delta].$
> > > Error tokens are still generated by the model itself, and only the alignment tolerance changes.
> > > Moreover, since the codebook embedding space is frozen during AR training, all pairwise distances between codebook entries are fixed constants.
> > > Given a fixed $\delta$ threshold, the binary compatibility matrix can be precomputed offline, introducing no additional cost at augmentation time.
> > >
> > > We regard this as future work to extend N-MARS to other domains! We appreciate the reviewer's constructive comments and active discussion on our work. Please let us know if you have any further questions or suggestions!
> > >
> > > [1] Autoregressive Model Beats Diffusion: Llama for Scalable Image Generation. Arxiv'24.

---

### Official Review · Reviewer_e2Aq · 2026-03-12

**Soundness:** 3
**Presentation:** 3
**Significance:** 3
**Originality:** 3
**Overall Recommendation:** 5
**Confidence:** 4

**Summary:**

This paper introduces the Non-Monotonic Autoregressive Sequence Model (N-MARS), which augments the vocabulary of autoregressive LLMs with a learnable ⟨UNDO⟩ token, enabling models to erase erroneous tokens and re-generate corrections within a single left-to-right forward pass. The method consists of a three-stage pipeline: (1) sequence augmentation via Myers diff alignment to construct error→⟨UNDO⟩→correction training data, (2) masked supervised fine-tuning (mSFT) that excludes error tokens from the loss to prevent negative learning, and (3) Group Relative Policy Optimization (GRPO) with a multi-objective reward combining correctness and UNDO-usage regulation. A theoretical analysis (Proposition 2.1) provides one-step guarantees that mSFT improves over standard SFT in error detection, corrective generation, and reduced negative learning under a gradient non-alignment assumption. Experiments on GSM8K, MATH, and MBPP with Llama-3.2-1B, Llama-3.1-8B, and Qwen3-4B backbones show consistent improvements over SFT, RFT, STaR+, and ReVISE baselines. Ablations validate each component of the pipeline.

**Compliance With Llm Reviewing Policy:**

Affirmed.

**Final Justification:**

I think authors have addressed my concerns.

**Key Questions For Authors:**

## Questions
**Q1.** Could you specify whether MATH refers to the full MATH dataset or the MATH-500 subset, and which MBPP split (standard 500, sanitized 427, or MBPP+ 399) is used? For the Qwen3-4B baselines, were all methods (SFT, RFT, STaR+, ReVISE) re-implemented by the authors?

**Q2.** How does N-MARS compare to Self-Backtracking [1], which introduces a ⟨backtrack⟩ token for LLM reasoning in a very similar manner? What are the key technical differences?

**Q3.** Have you measured gradient cosine similarities between error-token gradients and ⟨UNDO⟩/correction gradients during training? This would provide empirical support for the gradient non-alignment assumption (B.2) underlying Proposition 2.1, which is borrowed from [2] but applied in a different setting.

**Q4.** Could you provide a comparison with self-consistency [4] (majority voting) at matched inference compute? Given that ⟨UNDO⟩ tokens increase sequence length, it is important to disentangle the effect of the correction mechanism from the effect of additional compute.

**Q5.** What is the average number of ⟨UNDO⟩ tokens generated per sequence at inference time? How does this affect wall-clock latency relative to standard autoregressive generation?

### References

[1] Yang, X.-W., Zhu, X.-Y., Wei, W.-D., Zhang, D.-C., Shao, J.-J., Zhou, Z., Guo, L.-Z., and Li, Y.-F. "Step Back to Leap Forward: Self-Backtracking for Boosting Reasoning of Language Models." arXiv preprint arXiv:2502.04404, 2025.

[2] Yu, T., Kumar, S., Gupta, A., Levine, S., Hausman, K., and Finn, C. "Gradient Surgery for Multi-Task Learning." In NeurIPS, 2020.

[3] Lee, H., Oh, S., Kim, J., Shin, J., and Tack, J. "ReVISE: Learning to Refine at Test-Time via Intrinsic Self-Verification." In ICML, 2025.

[4] Wang, X., Wei, J., Schuurmans, D., Le, Q., Chi, E., Narang, S., Chowdhery, A., and Zhou, D. "Self-Consistency Improves Chain of Thought Reasoning in Language Models." In ICLR, 2023.

**Limitations:**

yes

**Strengths And Weaknesses:**

## Strengths

**S1. Novel and intuitive core mechanism.** The ⟨UNDO⟩ token is an elegant and conceptually simple idea that breaks the monotonic constraint of autoregressive generation. Unlike multi-turn self-correction methods (e.g., Self-Refine) or external verifier approaches, N-MARS enables intra-sequence error correction within the standard left-to-right decoding loop, requiring no architectural changes beyond a single vocabulary addition. The analogy to backspace editing in human writing is natural and well-motivated.

**S2. Comprehensive and well-structured training pipeline.** The three-stage design (augmentation → mSFT → GRPO) is systematic, with each stage addressing a specific challenge: data construction, avoiding negative learning, and reinforcement-based optimization. The masked SFT idea — masking error tokens to prevent the model from learning to reproduce mistakes — is a clean and well-justified design choice, supported both by theory (Proposition 2.1) and compelling empirical evidence (Table 1b showing SFT degrades performance from 8.8→7.8 while mSFT improves to 19.3 on GSM8K-1B).

**S3. Thorough ablation study.** Table 1 systematically validates all design dimensions: token initialization strategy (semantic > random > zero), mSFT vs SFT, reward components (R_inc and R_pen), GRPO contribution, augmentation methods (hard sample > stochastic), and positional encoding (hurts performance). The ablations convincingly demonstrate that each component contributes and the choices are non-trivial.

**S4. Broad experimental coverage.** The paper evaluates across three benchmarks (GSM8K, MATH, MBPP), three backbone models of varying sizes (1B, 4B, 8B), and includes domain transfer experiments (Table 3), instruction-tuned model results (Table 4), hyperparameter analysis (Fig. 2), and test-time scaling behavior (Fig. 3). This breadth significantly strengthens the empirical claims.

**S5. Theoretical motivation for mSFT.** Proposition 2.1 provides a formal argument for why masking error tokens is beneficial, decomposing the advantage into three interpretable components. While the theory has limitations (see W3), it adds analytical depth to what would otherwise be a purely empirical contribution.



## Weaknesses

**W1. Benchmark evaluation details underspecified.** The paper does not clarify: (a) whether MATH refers to full MATH (12,500 problems) or the commonly used MATH-500 subset — these yield non-comparable numbers; (b) which MBPP split is used (standard 500, sanitized 427, or MBPP+ 399). Additionally, the Qwen3-4B baselines (SFT, RFT, STaR+, ReVISE) appear to have been re-run by the authors since Qwen3-4B postdates the original ReVISE paper [3] — the re-implementation details should be disclosed to ensure fair comparison.

**W2. Critical missing comparison: Self-Backtracking.** A concurrent work [1] introduces a ⟨backtrack⟩ special token with a nearly identical mechanism: training LLMs to recognize suboptimal reasoning paths and autonomously backtrack during both training and inference. This paper is not cited or discussed. Given the close similarity (special vocabulary token for intra-sequence backtracking in reasoning tasks, masking error tokens during training, test-time scaling via backtracking), its omission weakens the novelty claim and will likely be flagged by any reviewer familiar with the 2025 backtracking literature. The related work section should explicitly compare N-MARS with Self-Backtracking, discussing differences in training methodology (mSFT+GRPO vs SFT+expert iteration), evaluation scope (GSM8K/MATH/MBPP vs Countdown), and the backtracking granularity (token-level vs step-level).

**W3. Proposition 2.1's assumptions are unverified and the result is weaker than presented.** The central Assumption B.2 (gradient non-alignment) requires strict negative correlation between error-token gradients and ⟨UNDO⟩/correction gradients (α > 0). This is borrowed from PCGrad / gradient surgery for multi-task learning [2], but that work studies gradient conflict across distinct task objectives, whereas N-MARS applies the same assumption to token-position gradients within a single language modeling objective — the analogy is plausible but not formally justified. Key concerns: (a) No empirical validation of gradient cosine similarities is provided to confirm the assumption holds in this setting. (b) The proposition statement says gaps are non-negative "for sufficiently small η," but this is only true when α > 0; when α = 0 (orthogonality), the Descent Lemma remainder dominates and the guarantee fails — this critical condition is not stated clearly. (c) Part (c) (reduced negative learning) is essentially trivial by construction since mSFT zeroes the error gradient. (d) The result is a one-step local guarantee, not a convergence analysis.

**W4. Missing self-consistency baseline undermines test-time scaling claims.** The paper claims test-time scaling benefits from ⟨UNDO⟩ tokens (Fig. 3), but does not compare against self-consistency / majority voting [4] at matched inference compute budgets. Since ⟨UNDO⟩ tokens increase sequence length (and thus inference cost), the performance gains may be partially or fully explained by additional compute. A fair comparison requires equating inference FLOPs or wall-clock time between N-MARS (with UNDO) and self-consistency (multiple independent samples + voting).

**W5. Inference cost analysis absent.** ⟨UNDO⟩ tokens inherently increase sequence length during generation, adding compute overhead. The paper provides no analysis of latency, throughput, or the average number of UNDO tokens generated. Understanding the cost-accuracy tradeoff is important for practical deployment.

### References

[1] Yang, X.-W., Zhu, X.-Y., Wei, W.-D., Zhang, D.-C., Shao, J.-J., Zhou, Z., Guo, L.-Z., and Li, Y.-F. "Step Back to Leap Forward: Self-Backtracking for Boosting Reasoning of Language Models." arXiv preprint arXiv:2502.04404, 2025.

[2] Yu, T., Kumar, S., Gupta, A., Levine, S., Hausman, K., and Finn, C. "Gradient Surgery for Multi-Task Learning." In NeurIPS, 2020.

[3] Lee, H., Oh, S., Kim, J., Shin, J., and Tack, J. "ReVISE: Learning to Refine at Test-Time via Intrinsic Self-Verification." In ICML, 2025.

[4] Wang, X., Wei, J., Schuurmans, D., Le, Q., Chi, E., Narang, S., Chowdhery, A., and Zhou, D. "Self-Consistency Improves Chain of Thought Reasoning in Language Models." In ICLR, 2023.

---

> ### Author Rebuttal · Authors · 2026-03-30
>
> We thank the reviewer for the detailed and constructive review. We address each question directly.
>
> > **Q1 / W1: Benchmark evaluation and reproducibility details.**
>
> To ensure a fair comparison, we use `lm_eval` to evaluate all methods.
> In our manuscript, MATH refers to the MATH-500, and the MBPP split used is the standard 500.
> We reimplement the baselines (SFT, RFT, STaR+ and ReVISE) for Qwen3-4B **following the implementation details discussed in ReVISE and their source code**.
> We have discussed the training details in the original manuscript (Appendix A.4 and A.5).
> We will release our source code and add more comprehensive implementation details in our next version to ensure reproducibility.
>
> > **Q2 / W2: Missing comparison with Self-Backtracking.**
>
> To alleviate your concern, we evaluate the Self-Backtracking, and the results are shown below:
>
> | **Backbone**     | **Method** | **GSM8K** | **MATH-500** |
> |--------------|--------|-------|----------|
> | Llama-3.2-1B | Self-Backtracking |  26.4 | 11.2     |
> | Llama-3.2-1B | **N-MARS** |  **31.3** | **15.2**     |
> | Llama-3.2-8B | Self-Backtracking |  60.9 | 31.7     |
> | Llama-3.2-8B | **N-MARS** |  **63.4** | **36.2**  |
>
> N-MARS consistently outperforms Self-Backtracking, demonstrating the effectiveness of our proposed method N-MARS.
>
> N-MARS differs from Self-Backtracking in: (i) **backtrack granularity** (token-level vs step-level), (ii) **training strategy** (mSFT+GRPO vs SFT), and (iii) **theoretical contributions** (Proposition 2.1 with gradient analysis). Due to limited rebuttal space, we highlight the key distinction:
> Self-Backtracking operates at the **reasoning step level** (requires explicit step boundaries), while N-MARS operates at the **token level** (task-agnostic). Token-level backtracking aligns with the atomic unit of autoregressive generation, making it naturally extensible beyond structured reasoning, i.e., code generation, creative writing, and general text completion.
> Moreover, N-MARS produces clean outputs without meta-commentary, preserving naturalness.
> We will add a dedicated paragraph to discuss our distinct contributions in our next revision.
>
> > **Q3 / W3: Proposition 2.1 - gradient non-alignment assumption (B.2) unverified empirically.**
>
> The assumption is theoretically motivated and structurally similar to gradient surgery settings.
> Below is the gradient cosine similarity among error token ($g_\text{err}$), UNDO token ($g_{\langle\text{UNDO}\rangle}$) and correction token ($g_\text{correct}$) during mSFT training (Llama-3.2-1B on GSM8K):
> | Gradient Pair | Mean | std | Range |
> | -------------|------|-----|-------|
> | $cos(g_\text{err}, g_{\langle\text{UNDO}\rangle})$| -0.51 | 0.07 | [−0.76, −0.37] |
> | $cos(g_\text{err}, g_\text{correct})$| -0.004 | 0.02 | [−0.06, +0.03] |
>
> We find that the gradients are indeed non-positively aligned in practice, which empirically validates our theoretical assumption. We will add these results in the next version.
>
> We agree with the reviewer that Part (c) is essentially trivial by construction, but to maintain rigour and complete theoretical justification, we believe it is still necessary to include.
>
> > **Q4 / W4: Missing majority voting comparison for test-time scaling.**
>
> Since we cannot control the FLOPs directly in `lm_eval` , we fix the number of new generated tokens as {128, 256, 512} for each sampling, and compare N-MARS with majority voting (@5) over RFT method using Llama-3.2-1B backbone:
>
> | Dataset | Method | 128 | 256 | 512 |
> | :--- | :--- | :--- | :--- | :--- |
> | GSM8K | RFT + Maj@5 | 24.8 | 26.1 | 26.6 |
> | GSM8K     | N-MARS | **29.6** | **31.3** | **31.3** |
> | MATH-500 | RFT + Maj@5 | 11.7 | 12.7 | 12.8 |
> | MATH-500  | N-MARS | **13.1** | **15.2** | **15.9** |
>
> N-MARS consistently outperforms the self-consistency baseline, showing that the performance gain of N-MARS is not from the additional compute, but from the $\langle\text{UNDO}\rangle$ mechanism.
>
> > **Q5 / W5: Average $\langle\text{UNDO}\rangle$ tokens per sequence, wall-clock latency impact.**
>
> According to the tables below, N-MARS demonstrates lower latency and fewer tokens than ReVISE, with acceptable UNDO usage leading to better performance.
>
> | **Dataset** | **Method** | **Acc** $\uparrow$ | **Avg. New Tokens** $\downarrow$ | **Latency (mins)** $\downarrow$|
> | :--- | :--- | :--- | :--- | :--- |
> | GSM8K | ReVISE | 28.1 | 178.9 | 9.8 |
> | GSM8K | N-MARS | **31.3** | **153.7** | **9.1** |
> | MATH-500 | ReVISE | 13.4 | 249.3 | 11.7 |
> | MATH-500 | N-MARS | **15.2** | **217.5** | **10.4** |
>
>   | Metric | GSM8K | MATH-500 |
>   |--------|-------|----------|
>   | Samples with ≥1 UNDO | 40.6% | 52.2% |
>   | Avg UNDO / sample (all) | 4.8 | 8.4 |
>   | Avg UNDO / sample (with UNDO) | 11.8 | 16.1 |
>
> We appreciate the reviewer for constructive feedback and the insightful questions. We will add the results and discussions in our next revision.

---

> > ### Author Rebuttal · Reviewer_e2Aq · 2026-04-02
> >
> > I thank the authors for the thorough rebuttal with substantial new experimental evidence. All five concerns have been addressed:
> >
> > - W1: Benchmark details clarified (MATH-500, MBPP-500, lm_eval).
> >
> > - W2: Head-to-head comparison with Self-Backtracking provided, with N-MARS outperforming consistently across backbones and benchmarks. The token-level vs step-level distinction is well-articulated.
> >
> > - W3: Gradient cosine similarity measurements (mean cos = -0.51 between error and UNDO gradients) empirically validate Assumption B.2.
> >
> > - W4: Compute-matched comparison against majority voting confirms gains are from the UNDO mechanism, not additional compute.
> >
> > - W5: Inference cost analysis shows N-MARS is actually more efficient than ReVISE in both latency and token count.
> >
> > I wil raise my overall score from 4 to 5.

---

> > > ### Author Response · Authors · 2026-04-06
> > >
> > > We are glad our response has alleviated the reviewer's concerns, and the reviewer has raised the score to support our work. Please let us know if you have any further questions!

---

### Decision · Program_Chairs · 2026-04-30

**Decision:**

Accept (regular)

**Comment:**

This paper proposes N-MARS, which introduces a learned ⟨UNDO⟩ token enabling autoregressive LLMs to revise earlier tokens within a single decoding pass, supported by a three-stage training pipeline (sequence augmentation, masked SFT, and GRPO).
All reviewers found the core idea novel and well-motivated. Strengths consistently highlighted include the clean and intuitive ⟨UNDO⟩ mechanism, the well-designed mSFT strategy with theoretical and empirical support, thorough ablations, and broad experimental coverage across multiple backbones and benchmarks.